

# Coupled Land Surface-Subsurface Hydrogeophysical Inverse Modeling to Estimate Soil Organic Content and explore associated Hydrological and Thermal Dynamics in an Arctic Tundra

Anh Phuong Tran[1], Baptiste Dafflon[1], Susan S. Hubbard[1]

[1]Climate & Ecosystems Division, Earth and Environmental Sciences Area, Lawrence National Berkeley Lab, Berkeley, California, CA 94720, USA

*Correspondence to*: Anh Phuong Tran (aptran@lbl.gov)

**Abstract**: Quantitative characterization of soil organic carbon (OC) content is essential due to its
significant impacts on surface–subsurface hydrological-thermal processes and microbial decomposition of OC, which both in turn are important for predicting carbon-climate feedbacks. While such quantification is particularly important in the vulnerable organic-rich Arctic region, it is challenging to achieve due to the general limitations of conventional core sampling and analysis methods, and to the extremely dynamic nature of hydrological-thermal processes associated with
annual freeze-thaw events. In this study, we develop and test an inversion scheme that can flexibly use single or multiple datasets, including soil water liquid, temperature and electrical resistivity data (ERT), to estimate the vertical distribution of OC content. We subsequently explore the control of OC on hydrological-thermal behavior. We employ the Community Land Model to simulate nonisothermal surface-subsurface hydrological dynamics from the bedrock to the top of
canopy, with consideration of land surface processes and ice/liquid water phase transitions. For inversion, we combine a deterministic and an adaptive Markov chain Monte Carlo (MCMC) optimization algorithm to estimate posterior distributions of desired model parameters. For hydrological-thermal to geophysical variable transformation, the simulated subsurface temperature, liquid and ice water content are explicitly linked to soil apparent resistivity via
petrophysical and geophysical models. We validate the developed scheme using different numerical experiments and evaluate the influence of measurement errors and benefit of joint inversion on the estimation of OC and other parameters. We also quantified the propagation of uncertainty from the estimated parameters to prediction of hydrological-thermal responses. We find that compared to inversion of single dataset (either temperature or liquid or apparent
resistivity), joint inversion of these datasets significantly reduces parameter uncertainty. We find that the joint inversion approach is able to estimate OC and sand content within the shallow active layer (0.3 m) with high reliability. Due to the small variations of temperature and moisture within



the shallow permafrost (0.6 m), the approach is unable to estimate OC with confidence. However, if the soil porosity is functionally related to the OC and mineral content, the uncertainty of OC estimate at this depth remarkably decreases. Our study documents the value of the new surface-subsurface, deterministic-stochastic inversion approach, as well as the benefit of including multiple types of data to estimate OC and associated hydrological-thermal dynamics.

## 1. Introduction

Soil organic carbon (OC) and its influence on terrestrial ecosystem feedbacks to global warming in permafrost regions is particularly important for the calculation of global carbon budget and prediction of future climate variation. Warmer air temperature leads to permafrost degradation, which is expected to enhance decomposition of huge pools of previously-frozen OC, releasing carbon dioxide and methane to the atmosphere, and enhancing global warming (Koven et al., 2011; Schaphoff et al., 2013; Schuur et al., 2015). In that context, accurate estimation of OC content stored in both active layer and permafrost is crucial for investigation of carbon stocks exposing for microbial decomposition.

Predictive understanding of ecosystem feedbacks to climate in permafrost regions requires quantitative knowledge of surface-subsurface hydrological-thermal dynamics, which in turn are strongly governed by the hydrological-thermal properties of soil OC (Jafarov and Schaefer, 2016). In particular, there are dramatic differences between thermal and hydraulic properties of OC and mineral soil, both of which typically co-exist in shallow permafrost systems. OC's thermal conductivity (e.g., $\lambda_{OC,dry} = 0.05$ W/mK) is significantly lower than that of mineral soil (e.g., $\lambda_{sand} = 8.4$ W/mK). By contrast, its heat capacity is higher than mineral soil. Considering hydrological properties, the hydraulic conductivity of OC is higher and its capillary pressure is smaller than mineral soil (Lawrence and Slater, 2008). In addition, while mineral soil porosity typically ranges from 0.4 to 0.6, the porosity of OC soil is usually greater than 0.8. Due to its low thermal conductivity, a top OC layer can behave as an insulator that reduces the magnitude of heat and energy exchange between the atmosphere and deeper soil (e.g., Hinzman et al., 1991; Rinke et al., 2008). Nicolsky et al. (2007) and Jafarov and Schaefer (2016) reported that inclusion of vertical OC content profile into land surface model can considerably improve prediction of subsurface moisture, temperature and carbon dynamics. However, our ability to measure or estimate the distribution of OC is currently challenging, which inhibits accurate model prediction.





OC content is usually measured from core samples, which are collected from field sites and then analyzed in the laboratory (e.g., Kern, 1994). While this method is relatively accurate, it is labor intensive and typically limited in spatial coverage. There have been few studies attempting to indirectly estimate OC content (or its hydrological-thermal properties) by minimizing the

difference between numerical simulations and measurements of soil temperature or soil moisture in an inversion framework. For example, Nicolsky et al. (2009) employed a variational data assimilation technique to estimate the thermal parameters and porosity of different OC and mineral soil layers. Atchley et al. (2015) used soil temperature data to calibrate the hydrological-thermal properties of moss, peat and mineral soils. However, these studies only used single dataset

for inversion. There has been no study that jointly uses multiple types of data to improve the OC content estimation.

Geophysical methods hold potential for characterizing the subsurface in permafrost regions as well as their associated physical, hydrological and thermal processes. Geophysical techniques offer an

advantage over conventional point measurement techniques because they provide spatially extensive information in a minimally invasive manner (e.g., Hubbard and Rubin, 2005). For example, Arcone et al. (1998) and Chen et al. (2016) used GPR to characterize the depth of permafrost table. Hinkel et al. (2001) used GPR to estimate thaw depth, to recognize ice wedges and ice lenses, and to locate the organic-mineral soil interface. Lewkowicz et al. (2011) and You

et al. (2013) employed ERT, ground temperature monitoring, frost table probing and coring to detect the permafrost depth. Hubbard et al. (2013) combined Lidar data with multiple geophysical (ERT, GPR, electromagnetic) and point measurements to characterize active-layer thickness and permafrost variability in a large area.

In spite of the potential benefits offered by geophysical data for characterizing permafrost systems,

geophysical inversion approaches typically suffer from several challenges. First, inversion methods are often ill-posed due to the fact that geophysical observables are sensitive to different soil properties. Secondly, inversion approaches often typically require petrophysical models to link the geophysical observables with the property of interest. Finally, there are differences between the geophysical support scale and the scale of the imaging target (Hubbard and Linde,

2011). In order to take advantage of information inherent in geophysical signatures yet minimize the non-uniqueness challenges described above, many recent studies have explored the value of



coupled hydrogeophysical inversion frameworks for estimating soil properties (e.g., Kowalsky et al., 2011; Johnson et al., 2009; Tran et al., 2016). In these studies, the hydrological and geophysical models are coupled together so that geophysical data are used to estimate soil properties that control the subsurface hydrological-thermal dynamics. However, coupled hydrogeophysical inversion approaches developed to date are not adequate for use in permafrost systems due to several gaps. Developed methods have only been applied to snow-free systems without consideration of the significant dynamics associated with the free-thaw transition. Developed coupled hydrogeophysical inversion approaches have also not yet incorporated surface-subsurface interactions (e.g., evapotranspiration, energy balance, plant water uptake). Finally, while a few studies have used Soil Vegetation Atmospheric Transfer (SVAT) models to qualitatively interpret geophysical data (e.g., McClymont et al., 2013), to date, no study has coupled SVAT and geophysical models and data to improve property estimation.

Building on recent advances in the use of electrical methods to characterize permafrost properties as well as coupled hydrogeophysical inversion approaches described above, this study focuses on the development of a hydrological-thermal-geophysical inverse approach to estimate OC and its control on hydrological-thermal responses in Arctic systems using single or multiple datasets. Our approach advances and couples several algorithms. We use a SVAT model known as Community Land Model (CLM4.5, Oleson et al., 2013) to simulate water, heat and energy exchange from the bedrock to the top of canopy. The model considers most of the land surface processes, ice/liquid phase change and surface-subsurface hydrological-thermal dynamics. For parameter estimation, we combined deterministic and stochastic optimization algorithms to concurrently obtain the best parameter estimates and their associated uncertainties. The deterministic optimization algorithm is employed to estimate the initial parameter set and covariance matrix of the proposal distribution. For the stochastic optimization, we used an advanced MCMC method known as Delayed rejection Adaptive Metropolis (DRAM, Haario et al., 2006). With this implementation together with the adaptive MCMC algorithm, we expect to obtain the posterior probability distributions (*pdf*s) of the desired model parameters more quickly than the traditional MCMC technique. For hydrological-thermal to geophysical transformation, we explicitly consider the dependence of the soil apparent resistivity on the soil ice/liquid water content and soil temperature via petrophysical and forward geophysical models.





This study advances capabilities to estimate and understand the controls of OC on hydrological and thermal properties through developing a hydrological-thermal-geophysical inversion scheme and exploring its potential to estimate the vertical distribution of OC and mineral content at several depths within a representative synthetic Artic soil column. Herein, we use synthetic studies to: 1) evaluate the relationship between the measurement error and uncertainties of parameter estimates, 2) examine the improvement in parameter estimation offered by including various datasets in the inversion, including electrical resistivity data, 3) investigate how OC estimation changes if the mineral and petrophysical parameters are unknown, 4) explore how parameter estimation changes when soil porosity functionally correlates with the OC and mineral content, and 5) investigate the uncertainty propagation from the OC and mineral content to the hydrological-thermal prediction.

The paper is organized as follows. Section 2 describes the development of the hydrological-thermal-geophysical inversion scheme. Section 3 analyzes and discusses the results of different synthetic experiments. Summary and concluding remarks are provided in Section 4.

## 2. Methodology

The joint hydrological-thermal-geophysical inversion scheme developed in this study is shown in Figure 1. Generally, the scheme includes two main components: 1) A forward coupled hydrological-thermal-geophysical model that generates the subsurface state variables (i.e., ice/liquid content and temperature), and then transforms these variables to the apparent resistivity using a set of petrophysical formulas and a forward electrical resistivity model (Figure 1a); 2) A combined deterministic-stochastic optimization algorithm to estimate the *pdf*s of desired model parameters ($\theta$), which include the OC content vertical profile (scenarios from 1 to 9), sand content vertical profile (scenarios 8-9) and petrophysical parameters (scenarios 8-9) (see Table 2) by minimizing the misfit between measured and simulated data. It is worth noting that the scheme is developed so that single (e.g., soil temperature or liquid water content or apparent resistivity) or multiple datasets can be used for inversion.

### 2.1. Hydrological-thermal model

In this study, we employed CLM4.5 model (hereafter referred to as 'CLM'), which can effectively simulate different land surface energy balance and surface-subsurface hydrological-thermal



processes (Oleson et al., 2013). CLM represents horizontal heterogeneity using multiple parallel soil/snow columns having different land use and plant function types. The lateral flow between the soil columns is not accounted for in CLM. The model simulates the freeze-thaw dynamics by considering two phases of water: liquid and ice. The rate of phase change depends on the energy

excess (for the ice to liquid transition) or deficit (for the liquid to ice transition) from the soil temperature to the freezing temperature. Given CLM's ability to simulate different hydrological-thermal processes in cold regions, we found it suitable for Arctic soil column simulations. The minimum requirements for the top boundary conditions in CLM include precipitation, incident solar, air temperature and wind speed. The land use and plant type information can be provided by

users or extracted from the available model database. CLM assumes that soil is a mixture of three soil types, namely, OC, sand and clay. It calculates the soil hydrological-thermal parameters based on the fractions of these soil types and their corresponding hydrological-thermal properties (see Appendix A for more detailed information on these relationships). In this study, we focus on estimating OC and mineral content and exploring their control on hydrological-thermal behavior

under freeze-thaw conditions.

For more detailed exploration of the vertical variability of subsurface properties and associated hydrological-thermal dynamics, we increased the default number of soil layers in the CLM from 15 to 32 layers, and defined the depth of layer $i^{th}$ ($z_i$) as:

$$z_i = 0.025\left(e^{0.17(i-0.2)} - 1\right). \tag{1}$$

Of these 32 layers, CLM assumes that the 5 bottom layers are bedrock layers; hydrological-thermal dynamics are simulated in the top 27 soil layers, while only thermal dynamics are simulated in the 5 bottom bedrock layers. Equation 1 was used to ensure that the layer thicknesses near soil surface are thinner than those near the bottom (as shown in Figure 3) in order to capture

the important hydrological and thermal dynamics at the topsoil active layers.

Moreover, in order to explore how the soil porosity influences the estimation of soil OC and sand content, we modified the CLM to consider two cases 1) the soil porosity profile was fixed and independent from the soil OC and sand content (see scenarios from 1 to 8 in Table 1), and 2) the

30 soil porosity was calculated from the OC and sand content as default in the CLM (see scenario 9 in Table 1) as below (Lawrence and Slater, 2008):



$$\Phi = \frac{(100 - \%OC)\Phi_{min} + \%OC\Phi_{OC}}{100}, \tag{2}$$

in which $\Phi$ is the soil porosity; $\%OC$ is the OC content in soil; $\Phi_{min}$ and $\Phi_{OC}$ are, the porosity of mineral and OC, respectively. In the CLM, the OC porosity is given as $\Phi_{OC} = 0.9$ and the mineral conductivity is calculated from sand fraction as

$$\Phi_{min} = 0.489 - 0.00126(\%\text{sand}), \tag{3}$$

To illustrate the dependencies between soil properties, Figure 2 shows the soil thermal conductivity, heat capacity and thermal diffusivity as a function of liquid saturation, OC and sand content. This figure was obtained from calculations using equations in Appendix A in which the soil porosity was considered in two cases: 1) fixing at 0.7 (Figures a, b, c), and 2) calculating from the OC and sand content (Figures d, e, f). The figure shows that the variation of soil thermal properties with respect to the OC content, sand content and liquid saturation is similar for both cases. When the OC fraction increases from 0 to 100%, the soil thermal conductivity decreases and the soil heat capacity increases. By contrast, higher sand fraction leads to higher thermal conductivity and lower heat capacity. These relationships are expected, given that OC has a considerably smaller thermal conductivity and a slightly higher heat capacity compared to sand. The figure also shows that both soil thermal conductivity and heat capacity significantly increase with increasing liquid saturation. This is also reasonable, as the thermal conductivity and heat capacity of liquid water are much higher than that of air. The thermal diffusivity is defined as the ratio between the thermal conductivity and heat capacity. The figure indicates that the diffusivity increases when the OC decreases and sand content increases.

Compared the two cases of soil porosity, the figure shows that when the soil porosity is defined as a function of OC and sand content, the soil thermal properties change more quickly in larger ranges with the variation of OC content, sand content and liquid saturation. It is because while the soil porosity is fixed at 0.7 in the first case (Figure a, b, c), it varies from 0.36 (when soil is 100% sand) to 0.9 (when soil is 100% OC) in the second case (Figure d, e, f). Because soil thermal properties strongly depends on soil porosity (see Equations A2 and A8 in Appendix A), together with the OC and sand content, the variation of porosity in the second case leads to quickly change of the soil thermal properties, and therefore, the subsurface hydrological-thermal dynamics.





### 2.2. Petrophysical and geophysical transformation

In our inverse scheme, we link the output of hydrological-thermal simulation described above (soil ice/liquid saturation and temperature) to soil electrical conductivity using the following petrophysical relationship:

$$\sigma = \phi^m (S_w^n \sigma_w + (\phi^{-m} - 1)\sigma_s), \qquad (4)$$

in which $\phi$ is the porosity; $S_w$ is the liquid water saturation in the pore space; $m$ and $n$ are the cementation and saturation indexes, respectively, and $\sigma_s$ is the soil electrical conduction, which was fixed at $\sigma_s$=0.005 S/m in this study (Table 1). The water electrical conductivity is calculated from the concentration of all ions in water as (Minsley et al., 2015):

$$\sigma_w = \sum_{i=1}^{i=n_{ion}} F_c \beta_i |z_i| C_i, \qquad (5)$$

in which $\beta_i$, and $z_i$ are the ionic mobility and valence of the $i^{th}$ ion, respectively. Similar to Minsley et al. (2015), we assumed that Na$^+$ and Cl$^-$ are two main ions in this synthetic study. $F_c$ is Faraday's constant. $C_i$ is the concentration of the $i^{th}$ ion, which depends on the ice/liquid fraction as:

$$C_i = C_{i\,(S_{fi}=0)} S_{fw}^{-\alpha}, \qquad (6)$$

in which $S_{fi}$ and $S_{fw}$ are, respectively, the fraction of ice and liquid in ice-liquid water ($S_{fi}+S_{fw}$=1); $\alpha$ varies from 0 to 1 which is the coefficient accounting for the reduction of soil water salinity when liquid saturation decreases. A larger $\alpha$ implies a larger increasing rate of ion concentration with decreasing liquid fraction. The concentrations of ions in the ice-free water ($C_{i\,(S_{fi}=0)}$) can be obtained from samples in the summer season. The values of $m$, $n$, $\sigma_s$, $\phi$, $F_c$, $\beta_i$, $C_i$ and $\alpha$ used in this synthetic study are presented in Table 1. Except for $m$, $n$ and $\sigma_s$, the other parameters were taken from Minsley et al. (2015). Of these parameters, $\alpha$ and $m$ are two most important parameters that control the relationship between geophysical and hydrological-thermal variables, we estimated them by inverting soil moisture, temperature and geophysical data in scenarios 8 and 9 (see Table 2).

The effect of soil temperature ($T$) on the soil electrical conductivity is formulated as:

$$\sigma_T = \sigma(0.018 * (T - 25) + 1). \qquad (7)$$

The linkage between soil electrical conductivity with the apparent resistivity is established by the electrical forward model. In this study, we used the forward model of the Boundless Electrical

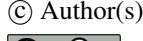



Resistivity Tomography (BERT) package, developed by Rücker et al. (2006), which numerically solves Poisson's equation using the finite element method in a three-dimensional arbitrary topography. For more detailed information on this model, we refer to Rücker et al. (2006).

**2.3. Stochastic and deterministic parameter estimation**

In this section, we present the combination of deterministic and stochastic optimization algorithms to estimate the model parameters $\theta$ and their uncertainties. The stochastic optimization algorithm relies on the Bayesian inference and DRAM MCMC technique. The deterministic optimization technique was used to approximate the initial set of model parameters and initial covariance

matrix of the proposal distribution for DRAM. Using this combined deterministic-stochastic algorithm, the posterior distributions of estimated parameters are expected to more rapidly obtain than using stochastic algorithm because the initial set of estimated parameters and proposal covariance matrix for MCMC simulations are inherited from deterministic optimization rather than from arbitrary values. In addition, the DRAM optimization algorithm allows us to

sequentially update the proposal covariance matrix and perform multiple tries to improve the acceptance rate. This algorithm has been proved to be more efficient than the commonly-used MCMC Metropolis-Hasting method.

**2.3.1. Bayesian inference**

In the stochastic parameter estimation, the objective is to find the posterior probability distribution $P(\theta|Y)$ of parameters $\theta$ conditioned on the measurements $Y$ from which we can extract the best-estimated parameters and their uncertainties. Based on Bayesian rule, this posterior distribution is formulated as follows:

$$p(\theta|Y) \propto p(\theta)p(Y|\theta), \qquad (8)$$

in which $p(\theta)$ is priori parameter distribution of parameter $\theta$ and $p(Y|\theta)$ is the likelihood function. Assuming that the error residuals are uncorrelated, the likelihood function can be written as:

$$p(Y|\theta) = \prod_{i=1}^{n} f_{y_i}(y_i|\theta), \qquad (9)$$

where $f_{y_i}(y_i|\theta)$ denotes the probability density function (*pdf*) of measurement $y_i$ at time $t_i$ given

the model parameters $\theta$. If we further assume the error residuals (difference between modeling and measurement) to be normally distributed, then $f_{y_i}(y_i|\theta)$ can be written as:





$$f_{y_i}(y_i|\theta) = \frac{1}{\sqrt{2\pi\sigma_i^2}} exp\left[-\frac{1}{2}\left(\frac{\hat{y}_i - y_i(\theta)}{\sigma_i^2}\right)^2\right], \tag{10}$$

and Equation 9 becomes:

$$p(Y|\theta) \propto p(\theta)\left(\frac{1}{\sqrt{2\pi}}\right)^n \prod_{i=1}^n \frac{1}{\sqrt{\sigma_i^2}} exp\left[-\frac{1}{2}\sum_{i=1}^n \left(\frac{\hat{y}_i - y_i(\theta)}{\sigma_i^2}\right)^2\right], \tag{11}$$

in which $y_i(\theta)$ is the model response at time $t_i$; $\sigma_i^2$ is the variance of measurement error at time $t_i$.

### 2.3.2. Delayed rejection adaptive Metropolis (DRAM) Markov Chain Monte Carlo method

Once posterior density distribution $p(\theta|Y)$ of the model parameters is defined, we need to determine its statistical properties (e.g., mean, covariance). However, due to the nonlinearity of the

dynamic model, it is usually difficult to analytically obtain these properties. In that respect, the Monte Carlo methods can be used to generate samples from this posterior distribution and then calculate these properties. We employed the DRAM method that was improved from the Metropolis-Hasting MCMC method for this purpose. Basically, this method is a combination of the adaptive Metropolis and delayed rejection algorithm and briefly presented as follows:

*Metropolis-Hasting:* Given the current parameter set $\theta_k$ at iteration $k^{th}$, the candidate for the next move ($\theta'_{k+1}$) from the current value is generated from a proposal distribution $q_1(\theta_k, \theta'_{k+1})$. The acceptance ratio is calculated as below:

$$\alpha_1(\theta'_{k+1}, \theta_k) = min\left(1, \frac{\pi(\theta'_{k+1})q_1(\theta'_{k+1}, \theta_k)}{\pi(\theta_k)q_1(\theta_k, \theta'_{k+1})}\right), \tag{12}$$

where $\pi(\theta)$ is the target distribution needed to approximate ($p(\theta|Y)$). The next sample moves to the candidate $\theta'_{k+1}$, $\theta_{k+1} = \theta'_{k+1}$ if $\alpha > u$ with $u$ as a random variable generated from uniform distribution $U(0,1)$. Otherwise, the candidate is rejected and the next sample stays at the current location, $\theta_{k+1} = \theta_k$ .

*Delayed rejection*: In delayed rejection, once the candidate is rejected, instead of staying at the current sample, a second ($\theta''_{k+1}$) try is proposed. The acceptance ratio for this try is:





$$\alpha_2(\theta''_{k+1}, \theta'_{k+1}, \theta_k) = min\left(1, \frac{\pi(\theta''_{k+1})q_1(\theta''_{k+1}, \theta'_{k+1})q_2(\theta''_{k+1}, \theta'_{k+1}, \theta_k)(1-\alpha_1(\theta''_{k+1}, \theta'_{k+1}))}{\pi(\theta_k)q_1(\theta_k, \theta'_{k+1})q_2(\theta_k, \theta'_{k+1}, \theta''_{k+1})(1-\alpha_1(\theta_k, \theta'_{k+1}))}\right), \quad (13)$$

If the second try is rejected, the third try can be generated and so on. The number of tries is specified by users.

*Adaptation*: One of the key limitations of the MCMC technique is the selection of the proposal distribution model. In adaptive Metropolis, the proposal distribution is assumed to be Gaussian centered at the current sample $N(\theta_k, C_k)$ with the covariance matrix $C_k$ adapted from the previous samples as:

$$C_t = s_d cov(\theta_0, \dots, \theta_k) + s_d \varepsilon I_d. \quad (14)$$

In Equation 14, $s_d$ is the scaling parameter that depends on the length ($d$) of the estimated parameter vector $\theta$, which is set to $s_d = 2.4^2/d$; $\varepsilon > 0$ is a very small constant to inhibit $C_k$ from becoming singular; and $I_d$ signifies the $d$-dimensional identity matrix.

The assessment of convergence of the MCMC chain is analyzed by Geweke's criterion, which
compares the means and variances of the beginning and end segments of the chain as below:

$$G_i = \frac{\overline{\theta_{i,a}} - \overline{\theta_{i,b}}}{\sqrt{\frac{s_{i,a}}{n_a} + \frac{s_{i,b}}{n_b}}}, \quad (15)$$

where $a$ denotes the beginning interval, which was selected as the first 10% of the chain, and where $b$ denotes the end interval, which was selected as the last 50% of the chain. $\overline{\theta_{i,a}}, \overline{\theta_{i,b}}$ are,
respectively, the mean of parameters $i^{th}$ of segments $a$ and $b$; $n_a$, $n_b$ are the number of samples in $a$ and $b$ segments; and $s_{i,a}$ and $s_{i,b}$ are their corresponding consistent spectral density estimates at zero frequency. The chain is considered to be converged if the $G_i$ score is within the 95% interval of the standard Gaussian distribution ($-1.96 \leq z_i \leq 1.96$).

**2.3.3. Deterministic optimization for approximating starting parameters and proposal distribution**

The speed of convergence of the MCMC optimization algorithm strongly depends on the initial model parameter $\theta_0$ and initial proposal distribution $q_1$. In order to reduce the number of iterations needed to obtain the posterior distribution of model parameters, we used the local optimization



Nelder-Mead Simplex Method to approximate the starting model parameter and initial covariance matrix of the proposal distribution. The starting point of the DRAM is the best-estimated set of model parameters obtained by the Nelder-Mead method. The covariance matrix of the proposal distribution is assumed to be similar to that of the model parameters, which are locally calculated at the optimal solution obtained by the Nelder-Mead method as below:

$$\boldsymbol{C}_q^0 = \frac{1}{n-m} (\boldsymbol{J'J})^{-1} \sum_{i=1}^{n} \left( \frac{\widehat{y}_i - y_i(\boldsymbol{\theta})}{\sigma_i^2} \right)^2,  \tag{16}$$

in which $\boldsymbol{C}_q^0$ denotes the initial covariance matrix of the proposal distribution $q_1$ and $\boldsymbol{J}$ is the Jacobian matrix, which is defined as below:

$$\boldsymbol{J} = \begin{bmatrix} \frac{\partial y_1(\boldsymbol{\theta})}{\partial \theta_1} & \cdots & \frac{\partial y_1(\boldsymbol{\theta})}{\partial \theta_m} \\ \vdots & \ddots & \vdots \\ \frac{\partial y_n(\boldsymbol{\theta})}{\partial \theta_1} & \cdots & \frac{\partial y_n(\boldsymbol{\theta})}{\partial \theta_m} \end{bmatrix}.  \tag{17}$$

The partial derivatives $\frac{\partial y_i(\theta)}{\partial \theta_j}$ ($i$=1, 2,…, $m$; $j$=1, 2,…, $n$) are calculated at the optimal solution of the Nelder-Mead Simplex Method. Because these derivatives cannot be solved using analytical methods, we approximated them using:

$$\frac{\partial y_i(\theta)}{\partial \theta_j} \approx \frac{y_i(\theta_1,…,\theta_j+\Delta\theta_j,…,\theta_m) - y_i(\theta_1,…,\theta_j,…,\theta_m)}{\Delta\theta_j},  \tag{18}$$

in which $\Delta\theta_j$ was set at 5% of the parameter $\theta_j$.

## 3. Results and discussion

### 3.1. Synthetic soil column description and boundary conditions




To test the value of the developed joint inversion approach under a range of conditions and assumptions, we performed several synthetic case studies using the numerical soil column illustrated in Figure 3. The synthetic column was developed to mimic typical soil and petrophysical properties associated with a high-centered polygon at the Next Generation Environmental Ecosystem, Artic (NGEE-Artic) tundra site near Barrow, Alaska (e.g., Hubbard et al., 2013). Figure 4 shows the intensive study transect. The transect is 35 m in length and covers three typical topography types in Barrow, namely, high-centered (HCP), flat-centered (FCP) and low-centered polygon (LCP). The thawing occurs during the growing season lasts from the beginning of June to the end of September. In the growing season while the LCP is fully saturated with standing water above soil surface, the HCP is relatively dry and unsaturated. ERT measurements were performed along the transect daily using Wenner-Schlumberger configuration with an electrode spacing of 0.5 m. Other measurements and conditions useful for our synthetic studies, including soil temperature, soil moisture, thaw depth, snow dynamics, and climate conditions were also measured (Dafflon et al., 2017). These data will be used for real application of the joint inversion scheme that is carrying on.

Soil properties and petrophysical information used for the synthetic studies are provided in Table 1. The "true" soil properties are based on the core sample analysis at the Barrow, AK site (Dafflon, personal communication) and the "true" petrophysical parameters were obtained from Minsley et al. (2015). It is worth noting that soil is represented in the CLM as a mixture of OC, sand and clay. As such, in order to estimate the soil mixture, it was sufficient for us to consider OC and sand content (in sand-clay mineral mixture) only.

We assumed that the vertical profiles of soil properties (porosity, OC and sand content) were constructed by interpolating their corresponding values at 4 depths $z_k$=0.15, 0.3, 0.6 and 1 m as below:

$$f_z = \begin{cases} f_1 & if \ z \leq z_1 \\ f_{k-1} + \frac{z-z_{k-1}}{z_k-z_{k-1}}(f_i - f_{i-1}) & if \ z_{k-1} \leq z \leq z_k \ (k = 2,3,4) \\ f_4 & if \ z \geq z_4 \end{cases} \quad (19)$$

where $f_z$ are the soil properties at depth $z$ and $f_k$ are the soil properties at the corresponding depth $z_k$=0.15, 0.3, 0.6 and 1 m. These depths were chosen to represent the vertical variations of OC content and soil porosity in the core samples collected at the NGEE-Arctic Barrow Alaska site.



We synthetically explored 9 scenarios using the newly developed inversion procedure. Descriptions of these scenarios are presented in Table 2. The objectives of these scenarios are as follows:

1. *Scenarios 1 and 2*: Evaluate the effect of measurement errors on uncertainties of soil OC estimates (using electrical resistivity data as an example).

2. *Scenarios 2, 3, 4, 5, 6 and 7*: Investigate the improvement in OC estimation gained by joint inversion of multiple hydrological, thermal and geophysical datasets compared with inversion of each single dataset.

3. *Scenarios 7 and 8*: Study how the parameter estimates and their associated uncertainties change if, in addition to OC content, sand content and petrophysical parameters are unknown.

4. *Scenario 8 and 9*: Explore the effect of soil porosity on the parameter estimation by comparing two cases: 1) Soil porosity profile is fixed and independent from the soil OC and sand content and 2) soil porosity is defined as a function of OC and sand content.

5. *Scenario 8*: Analyze the uncertainties, non-uniqueness, correlation and convergence of the inverse problem as well as evaluate the impact of parameter uncertainty on prediction of hydrological-thermal dynamics.

For all scenarios, we used daily time step meteorological forcing data (including air temperature, wind speed, short-wave and long-wave radiation and precipitation) collected at the Barrow site over a year period from 01/01/2013 to 31/12/2013, which includes a time period over which some of the soil and electrical datasets were also collected at the NGEE-Arctic site. The plant functional type information was obtained from the CLM database for the Artic region. The general approach that we followed to perform all synthetic scenarios is presented in Figure 5.

In order to account for the measurement errors, we assumed that the error distribution was Gaussian, and added error to synthetic data to obtain "noisy" synthetic data (hereafter referred to as observation data). Information on observation data is presented in Table 2. We set the standard deviation of ERT measurement error to 2% of synthetic data for scenario 1 (low measurement error) and to 5% for the other scenarios. We used a standard deviation of measurement errors of 0.5°C for soil temperature and 0.08 for soil liquid content. The standard deviation of liquid content




was set to be relatively high because we observed that the associated error measurements at Barrow were quite high. Observation data for inversion includes: 1) Apparent resistivity data at 7 most important time points during the year, which includes events such as thawing (day 163), summer growing season (days 185, 199 and 234), freeze-up (day 266) and frozen winter (days 292 and 312); 2) Soil temperature data at $z$=0.004, 0.16, 0.8, 1 and 2.4 m from day 49 to 365, which are most varying; 3) Liquid water content at depths 0.004, 0.05, 0.11, 0.2 m during the summer growing season (days 159 to 259).

For inversion, ranges were provided for unknown soil and petrophysical parameters based on Hubbard et al. (2013) and Dafflon et al. (2017) (Table 2). To minimize non-uniqueness in the inversion procedure, we ignored the small OC content at 1 m and the small sand content at 0.1 m. For scenarios from 1 to 7, we estimated OC content at $z$=0.1, 0.3 and 0.6 m (3 parameters). For scenarios 8 and 9, we estimated OC content at $z$=0.1, 0.3, 0.6 m, sand content at $z$=0.3, 0.6 and 1 m and petrophysical parameters $m$ and $\alpha$ (8 parameters). We assumed that there is no prior information on the estimated parameters. As a result, the prior distributions of OC and sand content were uniform distributed within their parameter ranges.

### 3.2. Simulation results

In order to estimate the posterior *pdf* of OC and sand content as well as petrophysical parameters, we generate 8000 samples for scenarios from 1 to 7 and 15000 samples for scenarios 8 and 9. The number of samples in scenarios 8 and 9 are larger because there is more number of estimated parameters in these scenarios. We selected the last 5000 samples having a Geweke's score less than 0.4 to construct the *pdfs* of these parameters. Their best estimates and associated uncertainties ($\sigma$) are, respectively, represented by the means and standard deviations of the samples and summarized in Table 2. Discussion and comparison of the scenarios are presented below.

### 3.2.1. Effect of measurement error on parameter uncertainty

The influence measurement error on the parameter uncertainties was considered by comparing scenario 1 and 2 using of apparent resistivity as an example. Scenario 1 assumed that the standard deviation of measurement error is 2% of synthetic apparent resistivity data (small measurement error), while this value for scenario 2 is 5% (large measurement error). For these two scenarios,



we estimated the OC content at $z$= 0.1, 0.3 and 0.6 m Figure 6 shows the probability functions of the OC at these depths. The figures indicate that the uncertainties of the estimated OC content at $z$= 0.1 and 0.3 m are considerably higher when the measurement error is larger. As shown in Table 2, when the measurement error of apparent resistivity increases from 2% to 5%, the standard deviation of the posterior OC samples increases three times from 0.2 to 0.6 at $z$=0.1 m and more than two times from 2.6 to 5.5 at $z$=0.2 m. At $z$=0.6 m, the OC content cannot be reliably obtained by both scenarios.

In order to investigate the non-uniqueness problem and the correlation between parameters, we estimated the misfit (sum of square differences) between the synthetic and sampled electrical resistivity data as a function of the OC content at $z$=0.1, 0.3 and 0.6 m for scenario 2 (Figure 7). This figure indicates that while the OC at $z$=0.1 m is well identified, the misfit negligibly changes when the OC content at $z$=0.6 m varies from 20 to 50%. This indicates that the apparent resistivity data is insensitive to OC content at $z$=0.6 m. This is reasonable, because this depth is within the permafrost (see Figure 13), where temperature insignificantly changes over time. Figure 7 also shows that there is a negative correlation between the OC content at $z$=0.3 and 0.6 m, which increases the uncertainties of OC estimates at both depths.

### 3.2.2. Influence of joint inversion of multiple data on parameter uncertainty

The effectiveness of the joint inversion of multiple datasets on the OC content estimation (at $z$=0.1, 0.3 and 0.6 m) was investigated by comparing results obtained from 6 scenarios that used 1) single apparent resistivity (scenario 2); 2) single temperature (scenario 3); 3) single liquid content (scenario 4); 4) temperature and apparent resistivity data (scenario 5); 5) liquid water content and apparent resistivity (scenario 6); and 6) liquid water content, temperature and apparent resistivity data (scenario 7). Figure 8 presents the probability distribution of the OC content of these scenarios. The figures indicate that joint inversion of apparent resistivity with either temperature and/or liquid content data significantly reduces the uncertainties of OC content at $z$=0.1 and 0.3 m. For example, compared to use single temperature dataset, the uncertainty of OC content reduces from 0.4 to 0.2 at $z$=0.1 m, and from 4.6 to 1.9 at $z$=0.3 m when joint using temperature, liquid content and apparent resistivity datasets. Finally, we found that even when all "observation" data are used, there is no improvement in the OC content estimate at $z$=0.6 m. These synthetic experiments suggest that given this depth is located within the permafrost (see Figure 13), the





apparent resistivity, liquid content and temperature data are insensitive to OC content.

### 3.2.3. Effect of mineral content and petrophysical parameters

In scenario 8, in addition to the OC content, we assumed that the sand content and petrophysical

parameters $m$ and $\alpha$ are unknown and estimated these parameters using the apparent resistivity, temperature and liquid content data. Their posterior probability distributions are presented in Figure 9. Similar to the previous scenarios, the OC content at $z$=0.1 and 0.3 m were obtained with small uncertainties ($\sigma_{OC}$ ($z$=0.1 m)=0.3, $\sigma_{OC}$ ($z$=0.3 m)=2). The sand content at $z$=0.3 and 1 m were also well estimated with uncertainties of 2.4 and 2, respectively. It is worth noting that regardless

of deep location, the sand content at 1 m is relatively well determined because at this depth the sand-clay mineral (92%) dominates the OC content (8%), and therefore, the hydrological-thermal data are relatively sensitive to this parameter. By contrast, the OC and sand content at $z$=0.6 m are unidentifiable with uncertainties up to 6.5 and 8.2, respectively. Finally, both of the petrophysical parameters $m$ and $\alpha$ are well specified but the parameter $\alpha$ is better obtained. This implies that $\alpha$ is

more sensitive to the apparent resistivity than to $m$.

The pairwise relationships between estimated parameters are shown in Figure 10. The figure indicates that the OC content at 0.1 m and petrophysical parameter $\alpha$ are the most reliably-estimated parameters, followed by the OC content at $z$=0.3 m, sand content at $z$=0.3 and 1 m, and

cementation index $m$. As for the correlation between parameters, the figure reveals that there is a strong positive correlation between the sand and OC content at $z$=0.6 m with a correlation coefficient of 0.86. This correlation and the insensitivity of the observations with their variations are two main reasons for the non-uniqueness of these two parameters. The pairs of $m$-$\alpha$ and the $OC_{z=0.1\ m}$ - $Sand_{z=0.3\ m}$ are also highly correlated, with correlation coefficients of 0.84 and 0.70,

respectively.

### 3.2.4. Effect of porosity dependence on OC and mineral content

In this section, we evaluate how the parameter uncertainties change when the porosity is determined as a function of the OC and mineral content. We perform this evaluation by comparing

scenarios 8 and 9. Table 2 shows that all information in scenario 9 is similar to that of scenario 8 except for the soil porosity. While the soil porosity in scenario 8 was fixed and independent from



the OC and sand content, it was calculated from the OC and sand content in scenario 9 as shown in Equations 2 and 3.

The *pdf*s of all estimated parameters for scenario 9 are shown in Figure 11. The figure indicates that compared to scenario 8, all uncertainties of sand and OC content in scenario 9 are smaller. Especially, the uncertainties of these parameters at $z$=0.6 m are significantly decrease from 6.5 to 3.8 (for OC content) and from 8.2 to 1.8 (for sand content). This can be explained by the fact that, in addition to thermal parameters, the OC and sand content in scenario 9 controls the soil porosity, which also influences the subsurface hydrological-thermal dynamics (see Figure 2). As a result, the temperature, liquid water and apparent resistivity data in this scenario are more sensitive to variations of OC and sand content than those in scenario 8. Consequently, these parameters are more identifiable. By contrast, the uncertainty of petrophysical parameters $\alpha$ and $m$ considerably increases from 0.002 to 0.022 (for $\alpha$) and from 0.042 to 0.066 (for $m$). This is because while it was fixed in scenario 8, the soil porosity depends on the OC and sand content in scenario 9. Therefore, the soil porosity in scenario 9 is also uncertain due to the uncertainties of the OC and sand content. Because the soil porosity, $\alpha$ and $m$ are closely correlated (see Equation 4), the uncertainty of soil porosity causes higher uncertainties of $\alpha$ and $m$.

### 3.2.5. Uncertainty propagation from parameters to the hydrological-thermal and thaw layer thickness prediction

In this section, we evaluate the impact of parameter uncertainties on the prediction of hydrological-thermal dynamics. Posterior samples of the OC, sand content and petrophysical parameters $m$ and $\alpha$ of scenario 8 were used for this analysis. Figure 12 compares the synthetic and estimated the soil temperature at $z$= 0.004, 0.16, 0.99 m and the liquid content at $z$=0.004, 0.05 and 0.11 m. The uncertainties of these predictions are shown in the figure as grey color regions with a confident interval of 95%. The figure indicates that the synthetic and estimated soil temperature and liquid content agree well with each other. However, the uncertainty of the soil temperature prediction is much smaller than that of the liquid content prediction. The average confidence intervals over the simulation period of the soil temperature prediction at $z$= 0.004, 0.16, 0.99 m are 2.3, 2.3 and 1.7% of the "observation" respectively, while these values for the soil liquid prediction at $z$=0.004, 0.05 and 0.11 m are 28.7, 16.1 and 12.9%, respectively. These differences can be explained by the high sensitivity to the OC and sand content and the larger



measurement errors of liquid water content compared to soil temperature.

Figure 13 compares the synthetic and estimated thaw depth over a year period using results obtained from scenario 8. Both estimated and synthetic cases show that soil water thaws around

middle of June and freezes again around the middle of September. The thaw depth varies from 0.2 m to 0.42 m. These results are compatible with our field survey data in Barrow (Dafflon et al., 2017), indicating that although this is a synthetic study, its simulation is relatively compatible with the Arctic tundra field measurements. As for the influence of parameter uncertainties on the thaw depth estimation, we observed that the parameter uncertainties only cause thaw depth variations

during warmest period of the year (beginning of August to middle of September). During other times of the year, the thaw depths corresponding to different parameter sets are similar.

The comparison between synthetic and predicted apparent resistivity data is presented in Figure 14. The figure shows that there is a very good agreement between them with no bias, which

implies that our inversion scheme converges to the best solution region. The confidence ranges corresponding to a level of 95% vary from 1.4 to 9.4% of the "observation" resistivity, which is suitable with the relative measurement error of 5%.

## 4. Summary and Conclusions

In this study, we developed and tested a surface-subsurface coupled hydrogeophysical inversion approach to estimate OC and its influence on hydrological-thermal behavior under Arctic freeze-thaw conditions. In our inversion scheme, the CLM model serves as a forward model to simulate the land-surface energy balance and surface-subsurface hydrological-thermal processes. The new scheme can jointly use different types of data for the inversion, including electrical resistivity data.

The dependence of soil electrical resistivity on temperature and ice/liquid content are explicitly accounted for within the inversion.

We developed an advanced optimization technique that combines the deterministic and stochastic optimization algorithms to obtain soil and petrophysical parameters and their associated

uncertainties. The stochastic optimization estimated the posterior distribution of model parameters by using the Bayesian inference and adaptive MCMC algorithm-DRAM. Meanwhile the deterministic optimization algorithm was used to approximate the starting set of model parameters



and the initial covariance matrix of the proposal distribution for the stochastic optimization, which helps to more quickly converge to the targeted posterior distribution.

We tested the inversion scheme using multiple synthetic experiments in 1-D soil column representative of the Artic tundra, where surface–subsurface hydrological and thermal regimes co-interact and are influenced by soil organic carbon and mineral content. The obtained results show that the new inversion approach well reproduced the synthetic data in all experiments. The shallow (upper 0.3 m) active layer OC and sand content and the petrophysical parameters can be reliably

obtained using ERT data and the inversion approach. When the soil porosity is fixed, the uncertainties of OC and sand content are very high in the permafrost section (0.6 m), even when soil temperature, liquid saturation and apparent resistivity data were jointly used in the estimation procedure. This suggests that when the porosity is fixed, the inversion approach is unable to significantly improve the estimation of OC within the permafrost, due to the small magnitude of

temporal variation of both temperature and soil moisture in that section. However, if the soil porosity is considered as a function of OC and sand content, the permafrost parameters can be reliably obtained because the variation of porosity with OC and sand content increases the sensitivity to ice/liquid water and temperature. Examining the relationship between measurement errors and parameter uncertainties, we found that the uncertainties of estimated parameters

increases with increasing measurement error. We also explored the improvement in parameter estimation when jointly using multiple data for the inversion. Compared to single dataset inversion (either temperature or soil moisture or electrical resistivity), joint inversion significantly reduces the uncertainties of estimated parameters, especially at 0.3 m depth. Finally, we quantified the influence of parameter uncertainties on the prediction of hydrological-thermal and thaw depth

dynamics. The obtained results show that the soil water liquid content prediction is more uncertain than the soil temperature and apparent resistivity predictions, due to its large measurement error and its high sensitivity to OC and sand content. The uncertainties in OC and sand content have an impact on the thaw depth estimation only during the warmest months of the year (August and September).

This study developed and tested a novel approach to estimate soil OC content using inverse modeling that can incorporate diverse hydrological, thermal and ERT datasets. In addition, the study also permitted exploration of surface-subsurface hydrological-thermal dynamics and spatiotemporal variations associated with free-thaw transitions. Given the importance of





characterizing OC as part of ecosystem and climate studies, the typical challenges associated with collecting and analyzing 'sufficient' core data to characterize the vertical and horizontal variability of OC associated with a field study site, and the increasing use of electrical resistivity data to characterize vertical, horizontal and temporal variability in shallow systems, the new inversion

5     approach offers significant potential for improved characterization of OC over field-relevant conditions and scales. It also offers significant potential for improving our understanding of hydrological-thermal behavior of naturally heterogeneous permafrost systems. The successful validation of this approach using 1D synthetic studies provides a foundation for extending it to 2D and applying it to real field data, which is research that is currently underway.

In this study, we concentrated on the impact of OC on the soil electrical resistivity via its hydrological-thermal properties. Recent studies indicated that the soil OC content largely influences ionic mobility, and therefore, changes the polarization and relaxation time of soil response to the applied current, which can be measured by spectral induced polarization (SIP)

(e.g., Schwartz, N., & Furman, 2015). As a result, our future study will explore the possibility to integrate SIP measurements into our coupled hydrological-thermal-geophysical inversion scheme. In that case, the OC content is linked to SIP measurements both by its hydrological-thermal and electrical polarization properties.

This study assumed that the salinity depends only on the water liquid/ice concentration. We are planning to integrate a solute transport model into hydrological-thermal model to account for the spatiotemporal dynamics of ions in water.

With advancements in data acquisition, the surface-subsurface hydrological-thermal dynamics

now can be monitored in real-time with a high temporal resolution using multiple above- and below-ground measurements including geophysical techniques. Our next stage is to expand the inversion scheme so that it can assimilates these data into hydrological-thermal models to improve the model prediction in real-time.

**Acknowledgments**

The Next-Generation Ecosystem Experiments (NGEE Arctic) project is supported by the Office of Biological and Environmental Research in the DOE Office of Science. This NGEE-Arctic research is supported through contract number DE-AC02-05CH11231 to Lawrence Berkeley National Laboratory. The authors would like to thank NGEE-Arctic PI Stan Wullscheleger



(ORNL) for support and Dr. Thomas Günther for providing the BERT codes.

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



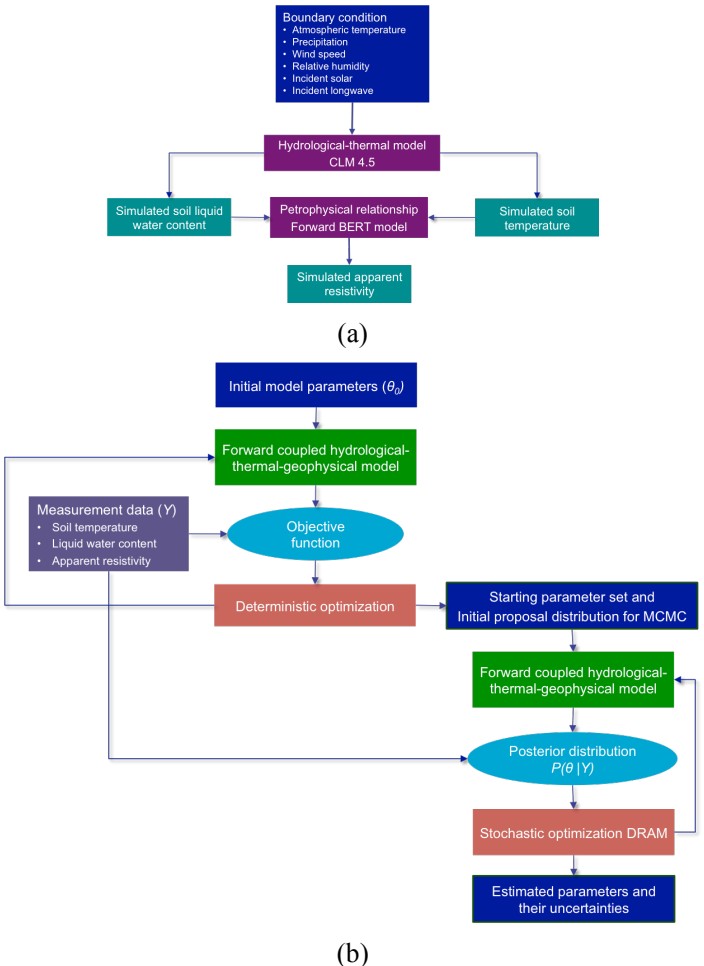

**Figure 1: (a)** Forward coupled hydrological-thermal-geophysical model that considers soil water liquid/ice content, temperature and apparent resistivity. **(b)** The two-stage inversion scheme combines deterministic and stochastic optimization algorithms to estimate the *pdf*s of desired model parameters (**θ**), which include the OC content (scenarios from 1 to 9), sand content (scenarios 8-9) and petrophysical parameters (scenarios 8-9) (see Table 2). The scheme permits to flexibly use single or multiple types of data for inversion. The forward coupled hydrological-thermal-geophysical model **(a)** is iteratively executed in both deterministic and stochastic inversion stages.





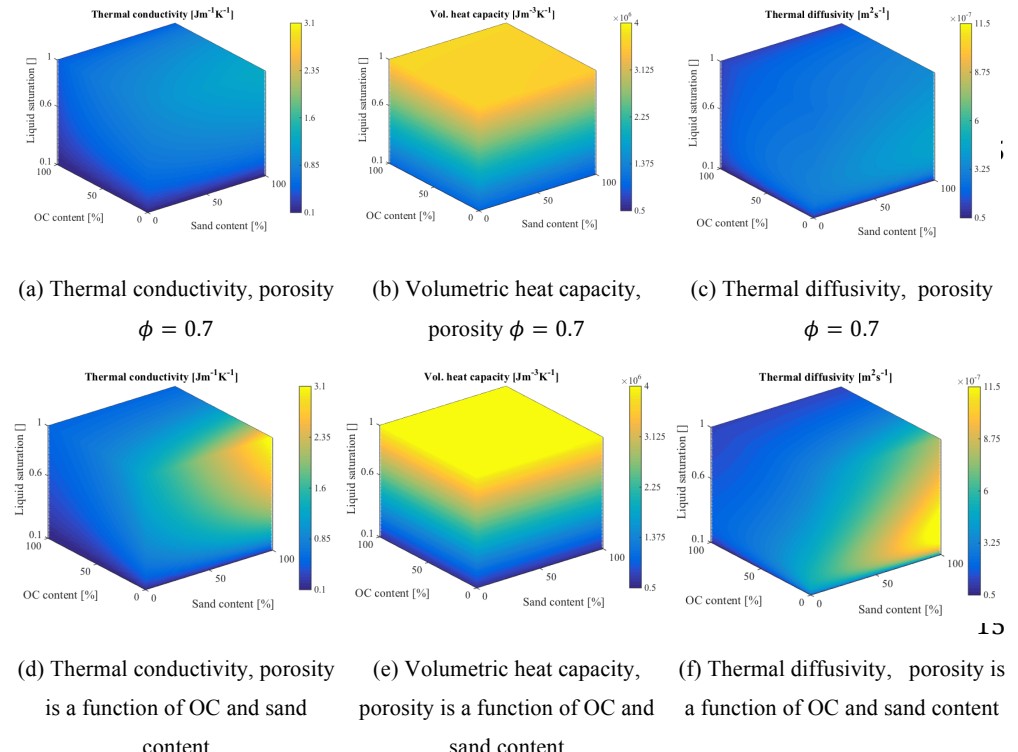

(a) Thermal conductivity, porosity $\phi = 0.7$

(b) Volumetric heat capacity, porosity $\phi = 0.7$

(c) Thermal diffusivity, porosity $\phi = 0.7$

(d) Thermal conductivity, porosity is a function of OC and sand content

(e) Volumetric heat capacity, porosity is a function of OC and sand content

(f) Thermal diffusivity, porosity is a function of OC and sand content

**Figure 2: Soil thermal conductivity (a, d), heat capacity (b, e) and thermal diffusivity (c, f) as a function of the liquid saturation, OC and sand content. The calculation for this figure was based on equations presented in Appendix A. The soil porosity was fixed at 0.7 for the top figures (a, b, c) and determined as a function of OC and sand content (Equations 2 and 3) for the bottom figures (d, e, f).**





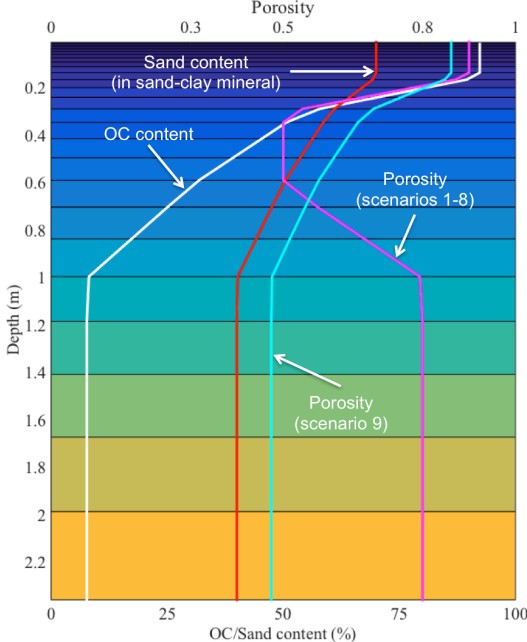

**Figure 3: 27 synthetic soil layers and soil properties (OC, sand content and porosity) for simulating hydrological and thermal dynamics. The 5 bottom bedrock layers are not shown in this figure. We assumed that the vertical profiles of soil properties are constructed by interpolating their corresponding values at $z$=0.1, 0.3, 0.6 and 1 m. For scenarios from 1 to 8, the soil porosity was fixed ($\Phi$=0.9, 0.5, 0.5, and 0.8 for $z$=0.1, 0.3, 0.6 and 1 m). For scenario 9, the soil porosity was calculated from the OC and sand content ($\Phi$=0.86, 0.67, 0.57, and 0.47 for $z$=0.1, 0.3, 0,6 and 1 m).**





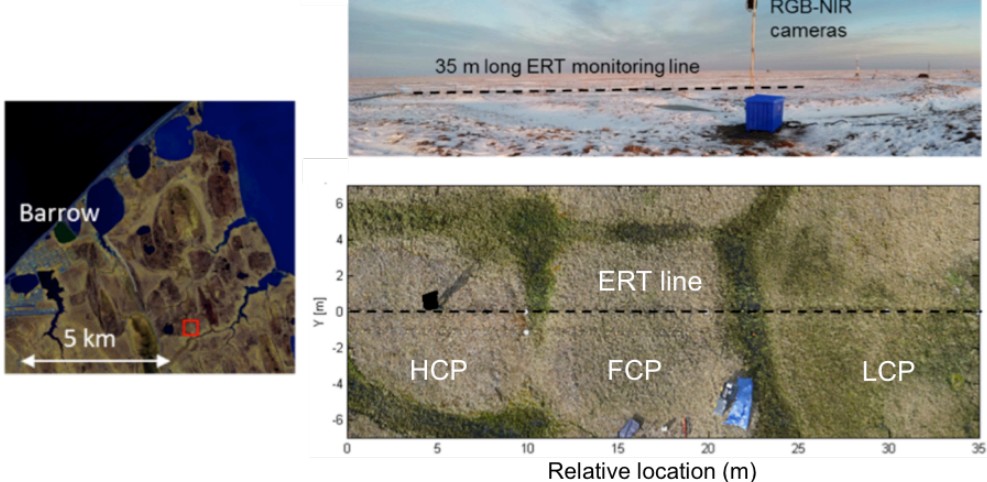

**Figure 4: (Left panel) Location of the study site (red square) near Barrow, Alaska, USA. (Top right panel) Image of the intensive ERT transect (dash line) and pole-mounted cameras, which monitor the land surface variability of the whole transect. (Bottom right panel) Aerial view of the ERT transect (dashed line), which covers different types of polygons, namely, high-centered polygon (HCP), flat-centered polygon (FCP) and low-centered polygon (LCP). (Modified from Dafflon et al., 2017).**





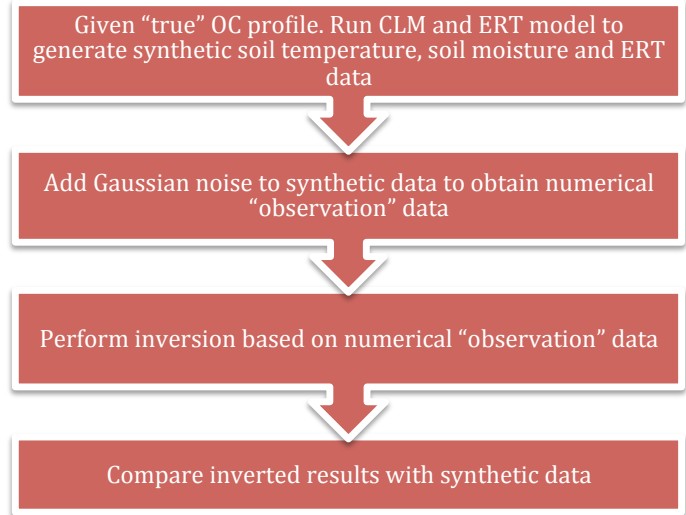

**Figure 5: General procedure used to perform the synthetic case studies.**




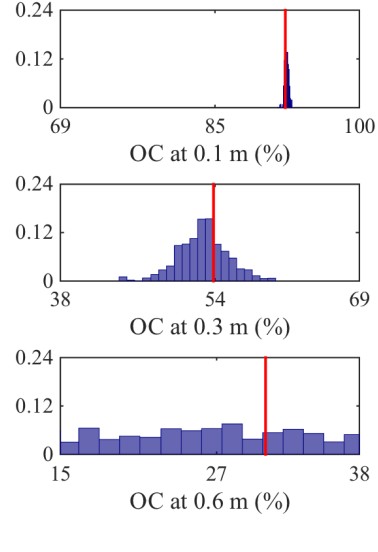
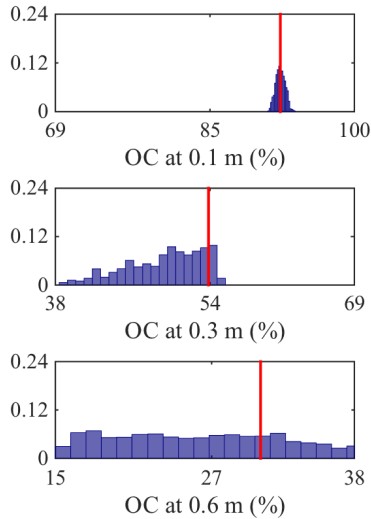

a) Scenario 1: ERT - 2 % error        b) Scenario 2: ERT - 5% Error

**Figure 6: The posterior probability of the OC content at z= 0.1, 0.3 and 0.6 m obtained by inverting apparent resistivity data with relative measurement error of 2% (a) and 5% (b). The red lines represent the "true" OC content.**




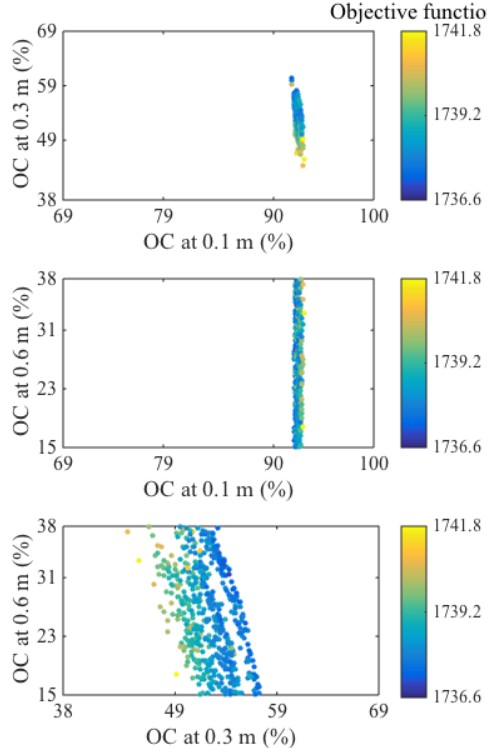

**Figure 7: Misfit (difference) between synthetic observations and MCMC sampling apparent resistivity data as a function of OC content at $z$= 0.1, 0.3 and 0.6 m for scenario 2.**





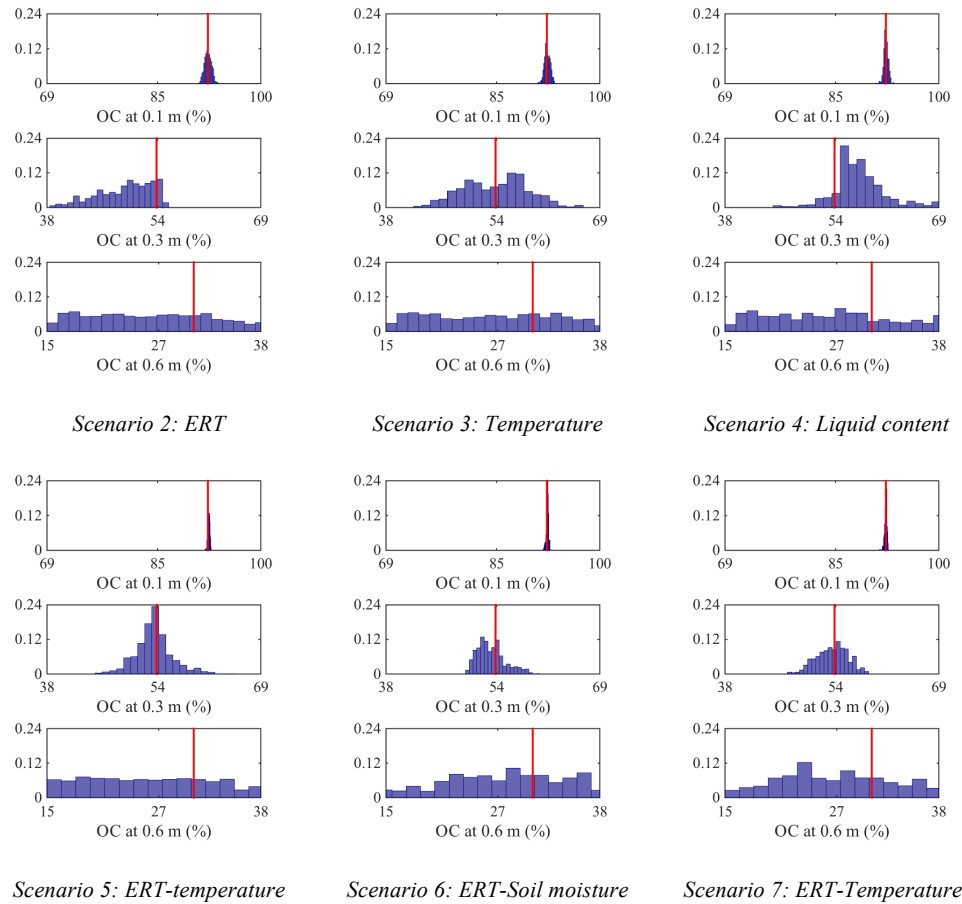

**Figure 8: The posterior probability of the OC content at z= 0.1, 0.3 and 0.6 m. These probability functions were constructed from 5000 MCMC samples. Measurement errors of apparent resistivity, temperature and liquid content data are 5%, 0.5 °C and 0.08, respectively. The red lines represent the "true" OC content.**





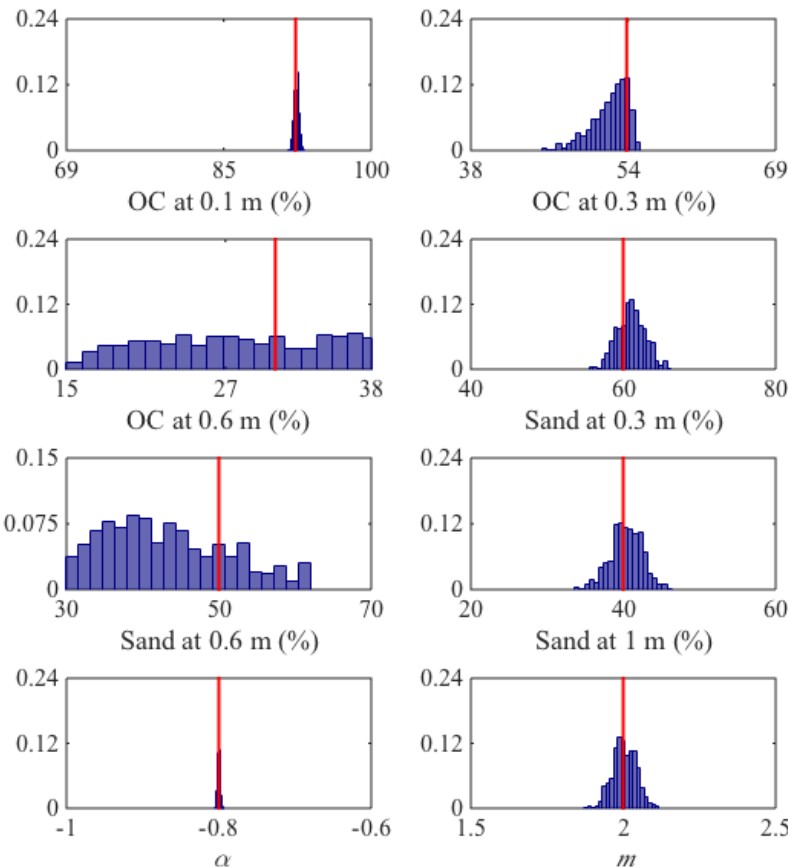

**Figure 9: Posterior probability of OC content at *z*=0.1, 0.3, 0.6 m, sand content at *z*=0.3, 0.6, 1 m and petrophysical parameters *m* and *α* for scenario 8. The sand content is the fraction of sand in the sand-clay mineral mixture. Soil porosity is fixed and independent from the OC and sand content.**



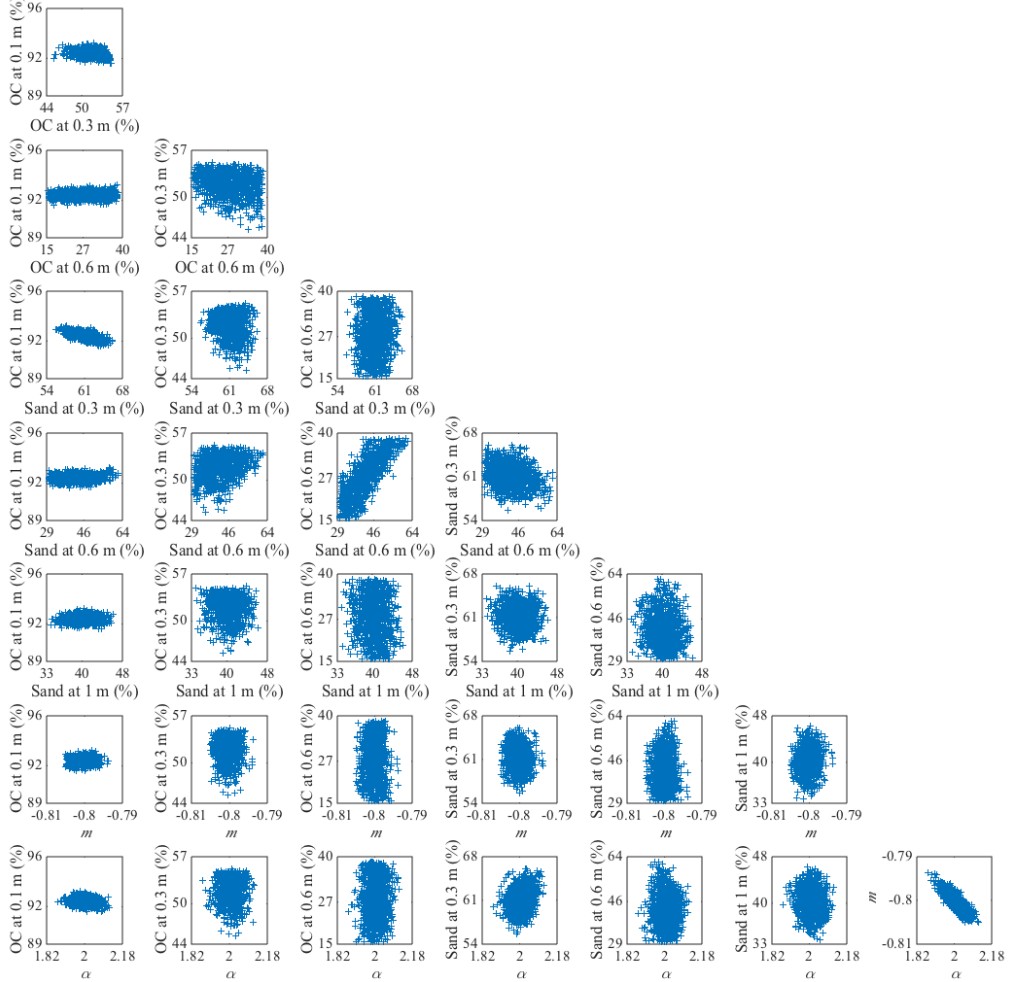

**Figure 10: Pairwise relationships between estimated parameters. The calculation was based on 3000 MCMC samples of scenario 8.**



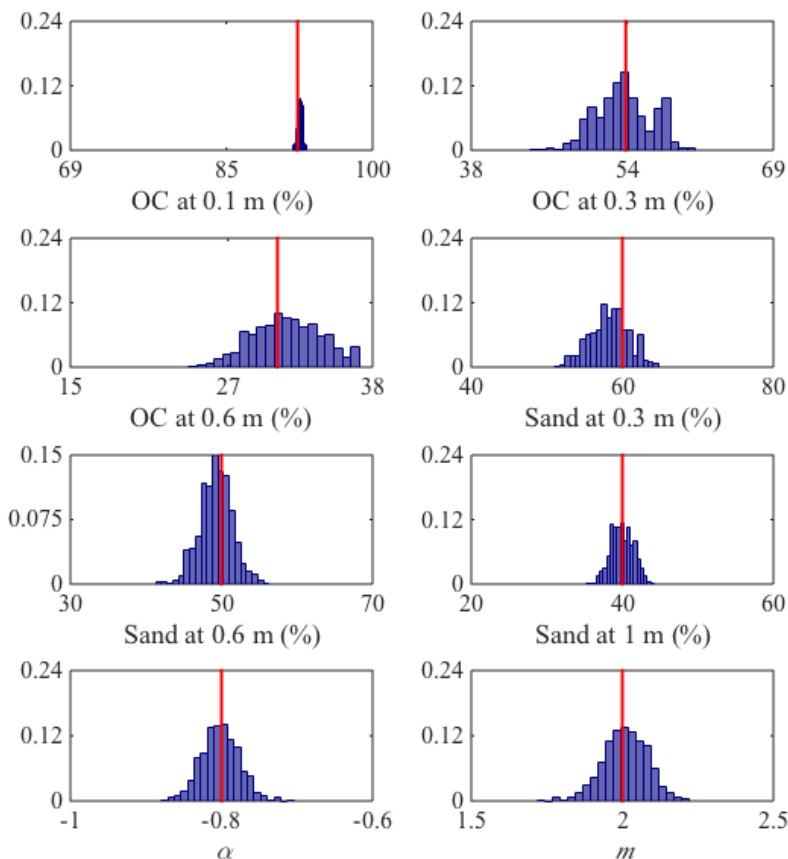

**Figure 11: Posterior probability of OC content at $z$=0.1, 0.3, 0.6 m, sand content at $z$=0.3, 0.6, 1 m and petrophysical parameters $m$ and $\alpha$ for scenario 9. The sand content is the fraction of sand in the sand-clay mineral mixture. Soil porosity is determined as a function of OC and sand content in the CLM.**



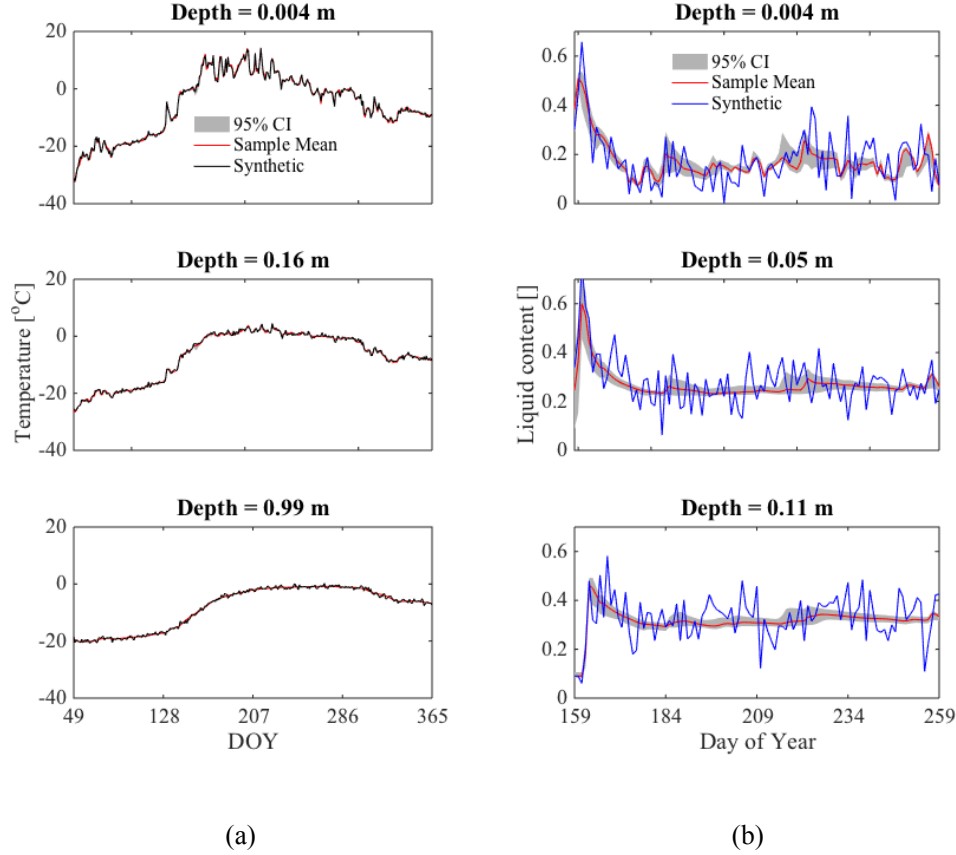

(a)                                              (b)

**Figure 12: Comparison of "observation" and predicted soil temperature at *z*=0.004, 0.16 and 0.99 m (a) and liquid water content at *z*=0.004, 0.05 and 0.11 m (b). The blue line denotes the synthetic data. The grey region represents the 95% confidence interval calculated from the posterior MCMC samples of scenario 8. The red line represents the mean of samples.**





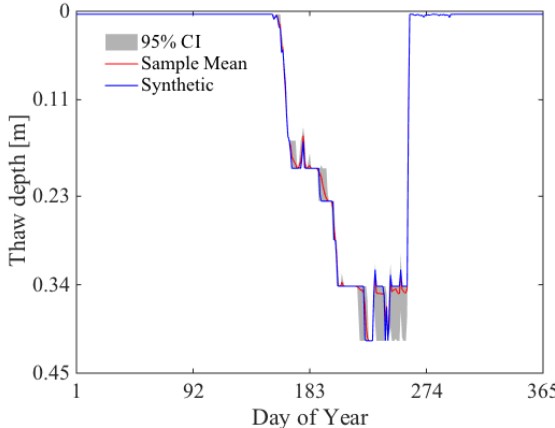

**Figure 13:** **Comparison between estimated and synthetic thaw depth over a year for scenario 8. The blue and red lines, respectively, represent the synthetic and estimated thaw depth. The grey region shows the confidence interval with a level of 95%.**



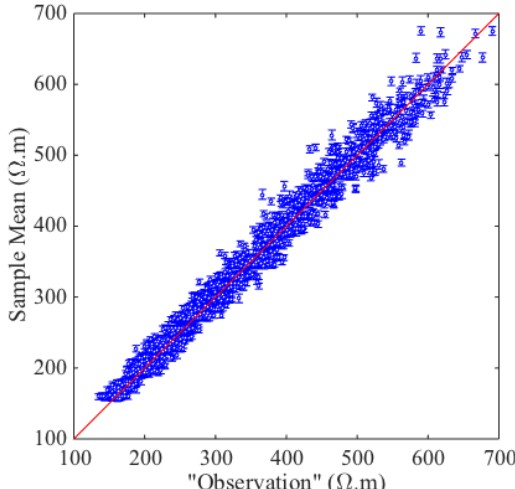

**Figure 14: Comparison between "observation" and predicted apparent resistivity. The red line demotes the 1:1 line. The vertical error bar on the blue symbols represents the confidence interval of the predicted apparent resistivity with a confidence level of 95%. The comparison was based on posterior samples of scenario 8.**



**Table 1: Petrophysical parameters and soil properties information used for synthetic simulation. Petrophysical parameters $F_c$, $C_{Na}^+$, $C_{Cl}^{-1}$, $\beta_{Na}^+$, $\beta_{Cl}^-$ are obtained from Minsley et al. (2015). Soil OC and sand content are based on the core sample analysis at the site near Barrow, AK. The sand content is the percentage of sand in the mineral mixture, which is calculated as 100-OC content. The soil porosity is independent from OC and sand content for scenarios from 1 to 8, while it is calculated from these properties in scenario 9.**

| Petrophysical parameter | | Soil properties | | | | |
|---|---|---|---|---|---|---|
| | | Depth (m) | $\phi$ (Scenarios 1-8) | $\phi$ (Scenario 9, calculated from OC and sand content) | OC content (%) | Sand content (in mineral mixture) (%) |
| $m$ | 2 | | | | | |
| $n$ | 1.3 | | | | | |
| $\sigma_s$ (Sm⁻¹) | 0.005 | | | | | |
| $F_c$ (Cmol⁻¹) | $9.6487\times10^4$ | | | | | |
| $C_{Na}^+=C_{Cl}^-$ (mol.m⁻³) | 4.28 | 0.1 | 0.9 | 0.86 | 92.3 | 70.0 |
| $\beta_{Na}^+$ (m²V⁻¹s⁻¹) | $5.8\times10^{-8}$ | 0.3 | 0.5 | 0.67 | 53.8 | 60.0 |
| $\beta_{Cl}^-$ (m²V⁻¹s⁻¹) | $7.9\times10^{-8}$ | 0.6 | 0.5 | 0.57 | 30.8 | 50.0 |
| $\alpha$ | -0.8 | 1 | 0.8 | 0.47 | 7.7 | 40.0 |





**Table 2: Information on parameter, "observation" data and inversion results for 6 scenarios. For scenarios from 1 to 7, the soil OC content (three parameters) at z=0.1, 0.3 and 0.6 m were estimated. For scenarios 8 and 9, the OC content at z=0.1, 0.3, 0.6 m, the sand content in the mineral mixture at z=0.3, 0.6 and 1 m and the petrophysical parameters m and α were estimated (8 parameters). The best estimated parameters and their uncertainties are represented by the mean and standard deviation of the MCMC samples.**

| Scenario | 1 | 2 | 3 | 4 | 5 | 6 | 7 | 8 | 9 |
|---|---|---|---|---|---|---|---|---|---|
| Data for inversion | Resistivity | Resistivity | Temperature | Liquid content | Resistivity and temperature | Resistivity and liquid content | Resistivity, temperature and liquid content | Resistivity, temperature and liquid content | Resistivity, temperature and liquid content |
| Measurement error | Resistivity: 2% | Resistivity: 5% | Temperature: 0.5°C | Liquid content: 0.08 | Resistivity: 5% Temperature: 0.5°C | Resistivity: 5% Liquid content: 0.08 | Resistivity: 5% Temperature: 0.5°C Liquid content: 0.08 | Resistivity: 5% Temperature: 0.5°C Liquid content: 0.08 | Resistivity: 5% Temperature: 0.5°C Liquid content: 0.08 |
| Porosity | Fixed (Value in Table 1) | | | | | | | Fixed (Values in Table 1) | Function of on OC, sand and clay content |
| Parameters | OC content at 0.1, 0.3 and 0.6 m | | | | | | | OC content at 0.1, 0.3, 0.6 m; Sand content at 0.3, 0.6 and 1 m; Petrophysical parameters $m$ and $\alpha$ | OC content at 0.1, 0.3, 0.6 m; Sand content at 0.3, 0.6 and 1 m; Petrophysical parameters $m$ and $\alpha$ |
| True value and Range for estimation | OC 0.1 m: *True value*: 92.3, *Range*: [69, 100]; OC 0.3 m: *True value*: 53.8, *Range*: [38, 69]; OC 0.6 m: *True value*: 30.8, *Range*: [15, 46] | | | | | | | OC 0.1 m: *True value*: 92.3, *Range*: [69, 100]; OC 0.3 m: *True value*: 53.8, *Range*: [38, 69]; OC 0.6 m: *True value*: 30.8, *Range*: [15, 46]; Sand 0.3 m: *True value*: 60, *Range*: [40, 80]; Sand 0.6 m: *True value*: 50, *Range*: [30, 70]; Sand 1.0 m: *True value*: 40, *Range*: [20, 50]; $\alpha$: *True value*: -0.8, *Range*: [-1, -0.5]; $m$: *True value*: 2, *Range*: [1.5, 2.5] | OC 0.1 m: *True value*: 92.3, *Range*: [69, 100]; OC 0.3 m: *True value*: 53.8, *Range*: [38, 69]; OC 0.6 m: *True value*: 30.8, *Range*: [15, 46]; Sand 0.3 m: *True value*: 60, *Range*: [40, 80]; Sand 0.6 m: *True value*: 50, *Range*: [30, 70]; Sand 1.0 m: *True value*: 40, *Range*: [20, 50]; $\alpha$: *True value*: -0.8, *Range*: [-1, -0.5]; $m$: *True value*: 2, *Range*: [1.5, 2.5] |
| Estimated OC content (%) | 0.1 m: 92.4±0.2; 0.3 m: 52.8±2.6; 0.6 m: 23±8.8 | 0.1 m: 92.1±0.6; 0.3 m: 53.7±5.5; 0.6 m: 26.9±6.5 | 0.1 m: 92.4±0.4; 0.3 m: 53.8±4.6; 0.6 m: 26.4±6.4 | 0.1 m: 92.3±0.3; 0.3 m: 53.4±3.7; 0.6 m: 26.1±6.4 | 0.1 m: 92.4±0.2; 0.3 m: 53.5±2.6; 0.6 m: 27.7±7.8 | 0.1 m: 92.3±0.2; 0.3 m: 53.4±2.2; 0.6 m: 28.0±6.9 | 0.1 m: 92.3±0.2; 0.3 m: 53.4±1.9; 0.6 m: 26.4±6.3 | 0.1 m: 92.3±0.3; 0.3 m: 52.6±2.0; 0.6 m: 31±6.5 | 0.1 m: 92.3±0.2; 0.3 m: 53.9±1.9; 0.6 m: 30.4±3.2 |
| Estimated sand content (%) | Not estimated | | | | | | | 0.3 m: 60.4±2.4; 0.6 m: 43.3±8.2; 1 m: 40.2±2.0 | 0.3 m: 60.1±1.8; 0.6 m: 49.5±1.8; 1 m: 39.5±0.9 |
| Estimated petrophysical parameters | Not estimated | | | | | | | $\alpha$: -0.801±0.002; $m$: 2.02±0.042 | $\alpha$: -0.806±0.022; $m$: 2.01±0.066 |





## Appendix A: Relationship between soil hydrological and thermal parameters and OC and mineral content

Soil thermal conductivity $\lambda$ is calculated as:

$$\lambda = \begin{cases} K_e\lambda_{sat} + (1-K_e)\lambda_{dry} & S_r > 1\times10^{-7} \\ \lambda_{dry} & S_r \leq 1\times10^{-7} \end{cases}, \tag{A1}$$

$$\lambda_{sat} = \lambda_s^{1-\theta_{sat}}\lambda_{liq}^{\frac{\theta_{liq}}{\theta_{liq}+\theta_{ice}}\theta_{sat}}\lambda_{ice}^{\left(1-\frac{\theta_{liq}}{\theta_{liq}+\theta_{ice}}\right)\theta_{sat}}, \tag{A2}$$

$$\lambda_{dry} = \lambda_{dry,om}f_{om} + \lambda_{dry,min}(1-f_{om}), \tag{A3}$$

$$K_e = \begin{cases} \frac{\theta_{liq}+\theta_{ice}}{\theta_{sat}} & \text{for forzen soil} \\ log\left(\frac{\theta_{liq}+\theta_{ice}}{\theta_{sat}}\right)+1 & \text{for unfrozen soil} \end{cases}, \tag{A4}$$

in which the saturated thermal conductivity of soil particles is calculated as:

$$\lambda_s = \lambda_{s,om}f_{om} + \lambda_{s,min}(1-f_{om}), \tag{A5}$$

in which the saturated and dry thermal conductivities of OC are $\lambda_{s,om} = 0.25$ and $\lambda_{dry,om} = 0.05$ Wm$^{-1}$K$^{-1}$, respectively. The saturated and dry thermal conductivity of minerals are calculated as:

$$\lambda_{s,min} = \frac{8.8(\%sand)+2.92(\%clay)}{\%sand+\%clay}, \tag{A6}$$

$$\lambda_{dry,min} = \frac{0.135\rho_d+64.7}{2700-0.947\rho_d}, \tag{A7}$$

where $\rho_d = 2700(1-\theta_{sat,min})$ is the bulk density of minerals.

The volumetric heat capacity is defined as:

$$c = c_s(1-\theta_{sat}) + \frac{w_{ice}}{\Delta z}C_{ice} + \frac{w_{liq}}{\Delta z}C_{liq}, \tag{A8}$$

with

$$c_{s,i} = (1-f_{om})c_{s,min} + f_{om}c_{s,om}, \tag{A9}$$

where $c_{s,min} = \frac{2.128(\%sand)+2.385(\%clay)}{\%sand+\%clay}\times10^6$ and $c_{s,om} = 2.5\times10^6$ Jm$^{-3}$K$^{-1}$

As for soil hydrological characteristics, soil matric potential and hydraulic conductivity are determined as:

$$\psi = \psi_{sat}\left(\frac{\theta}{\theta_{sat}}\right)^{-B}, \tag{A10}$$





$$
k_i = \begin{cases} 10^{-\Omega F_{ice}} k_{sat,i} \left[ \dfrac{0.5(\theta_i + \theta_{i+1})}{0.5(\theta_{i,sat} + \theta_{i+1,sat})} \right]^{2B_i+3} \\[2em] 10^{-\Omega F_{ice}} k_{sat,i} \left( \dfrac{\theta_i}{\theta_{i,sat}} \right)^{2B_i+3} \end{cases} ,
\tag{A11}
$$

with

$$
\psi_{sat} = (1 - f_{om})\psi_{sat,min} + f_{om}\psi_{sat,om},
\tag{A12}
$$

where the saturated matric potential of organic matter $\psi_{sat,om} = -10.3$ mm and the saturated mineral

5   matric potential is calculated as:

$$
\psi_{sat,min} = -10 \times 10^{1.88 - 0.0131(\%sand)},
\tag{A13}
$$

$$
B_i = (1 - f_{om,i})B_{min,i} + f_{om,i}B_{om,i} ,
\tag{A14}
$$

with $B_{om,i} = 2.7$.