# Peer review of "Coupled Land Surface-Subsurface Hydrogeophysical Inverse Modeling to Estimate Soil Organic Carbon Content and explore associated Hydrological and Thermal Dynamics in an Arctic Tundra"

_The Cryosphere, 2017_

## Referee Comment (RC1) · Anonymous Referee #1 · 9 Mar 2017

Review of manuscript tc-2017-1 "Coupled Land Surface-Subsurface Hydrogeophysical Inverse Modeling to Estimate Soil Organic Content and explore associated Hydrological and Thermal Dynamics in an Arctic Tundra " by A.P. Tran et al.

General comments:
This manuscript tests a coupled hydrological/geophysical inversion scheme combining the Community Land Model (CLM) and the Boundless Electrical Resistivity Tomography (BERT) model for estimating organic carbon content as well as hydraulic and

thermal soil parameters at an Arctic permafrost site. The inversion is conducted by combining two Markov chain Monte Carlo algorithms to infer uncertainties of the esti-mated model parameters. The modeling exercises are exclusively based on synthetic simulations where a set of scenarios for predicting the model parameters and the re-lated uncertainties is investigated. The manuscript is very well written and even though it is solely based on synthetic simulations, it fits well into the scope of The Cryosphere. The applied inversion scheme is at the forefront of algorithms applied in the field of hydrogeophysical inversion so far, e.g., also considering uncertainties. While being ap-plied in hydrological research rather frequently, to the reviewer's knowledge, it is one of the first being applied in a permafrost modeling study. The research question (es-timating soil organic carbon content under freeze/thaw conditions) is challenging and justifies to test the approach based on synthetic simulations. I have one major com-ment that I would like to see elaborated in a revised version of the manuscript. Then I am looking forward to seeing the paper published in The Cryosphere.

Major comment:
While I am very excited by the inversion approach, I am missing a derivation of the relationship between OC and the measured state variables (apparent electrical resistivity, soil moisture, soil temperature) that should be the basis for a successful, hence, a related section should be added to the revised manuscript including the respective references. In that context I am also somewhat disappointed about the selection of scenarios chosen for testing the inverse parameter estimation as well as the related discussion because I am often missing the physical basis (= discussion of relationships between different parameters or parameters and models) – please see also specific comments below.

Specific comments:

P 1, L 2: Better use "Soil Organic Carbon Content" here.

P 1, L 17: This already relates to my major comment above but even more to the description of the physical relationships later on in the manuscript: In order to be able to estimate OC content, there must be a physical relationship between liquid water content, temperature, apparent electrical resistivity and OC which needs to be elaborated and explained clearly and which should also be the basis for defining the scenarios.

P 1, L 20: Please provide examples for land surface processes ("such as...")

P 1, L 24: I would prefer to use "liquid water content and ice content" here.

P 2, L 22-25: Please add references and values for all listed properties here.

P 2, L 30: Please correct "into a land surface model"

P 3, L 9: Please correct "used a single dataset"

P 3, L 14-26: There are also numerous studies from Europe and Asia that use geophysics to study PF processes which should be honored in this short review as well.

P 4, L 10: Please correct "freeze-thaw", however freeze-thaw does not necessarily require the presence of snow

P 4, L 15: Which property? Please clarify.

P 4, L 32-33: Here the authors refer to the dependence of apparent resistivity to ice/liquid water content and soil temperature. I completely agree. But where is OC?

P 5, L 25: Here the authors use theta as variable for the parameters. In the appendix the same variable is used for soil moisture. I suggest to choose another variable for the parameters as most of the readers of TC will be used to theta as variable for soil moisture.

P 6, L 26: Please number figures in the order they appear in the manuscript.

P 6, L 27: I think this should read "in the topsoil active layers".

P 7, L 8 to end of section: I suggest to move this part of the section to the results

section. I think the important message from this exercise is that porosity needs to be considered in the model as soon as OC and mineral content are distinguished. I suggest to draw this conclusion and design or select the scenarios accordingly.

P 7, L 24: Replace "the figure" by "Figure 2"

P 7, L 29: Please correct "depend"

P 7, L 30: Please correct "to a quick change"

P 8, L 6: The authors chose to use Archie's Law here (please add reference). I am not an expert in ERT analysis but is it also applicable for soils with high OC? If yes, the relationship to OC would be in the porosity and the soil electric conduction (OC being a volume fraction of soil matrix then). How does Archie's Law deal with ice content (reduced porosity with ice in fact being a part of the soil matrix now)? As far as I see, ice content is only considered to calculate pore water conductivity but no changes in porosity. I would like to ask the authors to elaborate on that in this section.

P 8, L 27: Does this formula also apply to organic rich soils? Pleas add a reference.

P 9, L 12-16: Long sentence and difficult to understand. Can you split it into two?

P 13, L 3-6: This information is not essential for the manuscript and can be removed. (L5: Please correct "used".)

P 13, L 16: This is not so relevant for the synthetic study but probably for real-world cases to be carried out in future: As we are dealing with soil layers here, is it really reasonable to interpolate the measurements? Often we observe distinct soil horizons and also very distinct "jumps" in soil properties at layer boundaries and more constant properties along the different soil layers. Another aspect: how would ERT measurements have to be inverted considering gradients in layer properties? Please rethink whether interpolation is reasonable here.

P 13-14: Scenarios: As already stated in the general comments I am a little wondering

about the chosen scenarios. It would be helpful if to each set of scenarios a short explanation would be added why exactly these scenarios were chosen. In that context, I have the following questions which I would like to ask the authors to elaborate in the revised manuscript: Where is the relationship between OC and apparent resistivity in scenarios 1 and 2? Maybe this requires further explanation but as far as I see, in Archie's Law (eq. 4) OC primarily contributes to sigma via porosity and the soil's electrical conduction (which is fixed in this case however). So you basically vary a parameter in Archie's Law (or the BERT part of the scheme) whereas the "true" OC is modeled in CLM. How does that work? Is it reasonable to use electrical resistivity only as target variable? P 16, L 5-7: "This indicates that the apparent resistivity data is insensitive to OC content at z=0.6 m. This is reasonable, because this depth is within the permafrost (see Figure 12), where temperature insignificantly changes over time." > so apparent electrical resistivity depends on temperature but not on OC?

P 14, L 26-31: It would be nice to have the time series plots (cf Figure 12) already here including the apparent electrical resistivity data. Why didn't the authors decide to use the same time span for temperature and liquid water content? The changes in moisture and temperature (expept for layer 1) are rather low which I presume to be the main reason for the large uncertainties discussed later on. Hence the set of measurements is not really ideal for testing the inversion scheme. In order to obtain good parameter fits, the ranges in state variables where the model is fitted to should be large. Unfortunately this applies then to all tested scenarios.

P 14, L 26: I guess, this should be figure 11?

P 15, L 8: please correct: "uniformly"

P 15, L 15: Please correct "…8 and 9 is larger…"

P 15, L 20: Please correct: "influence of measurement error"

P 15, L 21: delete "of"

P 16, L 21: Please correct: "jointly"

P 16, L 23-25: "These synthetic experiments suggest that given this depth is located within the permafrost (see Figure 12), the apparent resistivity, liquid content and temperature data are in-sensitive to OC content." Why? Please explain. I presume the uncertainties are so large because there are almost no changes in state varibles the model is fitted to.

P17, L 9-17: I do not understand Figure 9. Why is ist reasonable to test correlations between all estimated parameters? What determines when a paramter is reliably estimated?

P17, L 29: remove "are"

P 17, L 31: So do "thermal parameters" mean ice content here?

P 18, L 18: Please correct "confidence interval"

P 18, L 26 to end of paragraph: Figure 12 is missing in manuscript.

P 19, L 30: Please correct "in a 1-D soil column"

P 20, L 5-7: Again: Here the authors relate the large uncertainty to the missing range in tempera-ture and moisture. However, again: Which are the properties that are related to OC? It's neither directly temperature nor moisture.

P 9, L 9-10: This sentence needs a more detailed explanation. Which property influences which state variable in which way?

P 20, L 13-15: Also here a more detailed conclusion would be helpful: In which way does the joint inversion help to constrain the model? How to the various measurements and models contribute here?

P 20, L 17-19: ... and the small range in "measured" soil moisture states.

P 21, L 1-2: This information is not relevant for the study and should be removed.

P 21, L 4-11: Also not relevant here – please remove.

References: Please check references for typos, also check that the European "Umlaute" are in-cluded in names (e.g. Etzelmüller)

P 22, L 13: only cite papers which are at least accepted

Appendix A: Definitions for most oft he parameters are missing.
* * *

---

## Referee Comment (RC2) · Anonymous Referee #2 · 20 Apr 2017

I enjoyed reading your paper and the authors of the paper describe an interesting combination of coupled hydrogeophysical inversion using also thermal properties. The investigation of uncertainties and to analyze the influence of different data sets to improve the results is a very interesting topic and highly important for coupled inversions. Coupled hydrogeophysical inversions have been widely used in the last years, but the extension to thermal parameters is still rare. Generally, I think the paper is well written and all the important steps are nicely explained. I have some minor comments that

could help to make same things clearer and improve the manuscript. After addressing them I recommend publication.

1. Introduction: Please discuss in more detail coupled hydrogeophysical inversion in terms of other geophysical methods. What are the benefits of ERT compare to GPR or seismic and why did you prefer this for you study. Mention the resolution of different methods and what are the limitations of what can be obtained.

2. Regarding the ERT data: a. Page 13: It would be nice to show also one ERT transect from the measured data and indicate the defined boundaries and structures in there. b. Please give more information of the ERT and the inversion. How reliable are the ERT results at a depth of 0.1m when using a spacing of 0.5m? c. Page 14, last paragraph: Considering measurement errors are highly important, but did you also consider uncertainties of the actually layer thicknesses obtained by the ERT? d. Page 15: Why do you use just 7 data set of the ERT, when you have data available for every day?

Technical corrections: The manuscript is very detailed, which is generally very good, but please try to remove unnecessary sentences to shorten the text and to better concentrate on the results. Avoid sentences like "Figure XX shows….". I listed here some examples that could shorten the text. Please check this for all the location where figures and tables are introduced. For example: e. Page 17: last paragraph. The three sentences can easily be combined to one. f. Page 18 starting line 4: First sentence not necessary and combine with second sentence. g. Page 19, second paragraph: Rewrite to "The comparison between synthetic and predicted apparent resistivity data (Figure 14) shows that there is a very good agreement between them with no bias…...

Congratulations on a very nice job!
* * *

---

## Author Comment (AC1) · 14 May 2017

We would like to thank the reviewer for their evaluation of our study and their detailed and constructive comments, which definitely helped to improve our paper.

General comments:

This manuscript tests a coupled hydrological/geophysical inversion scheme combining the Community Land Model (CLM) and the Boundless Electrical Resistivity Tomog-

raphy (BERT) model for estimating organic carbon content as well as hydraulic and thermal soil parameters at an Arctic permafrost site. The inversion is conducted by combining two Markov chain Monte Carlo algorithms to infer uncertainties of the estimated model parameters. The modeling exercises are exclusively based on synthetic simulations where a set of scenarios for predicting the model parameters and the related uncertainties is investigated. The manuscript is very well written and even though it is solely based on synthetic simulations, it fits well into the scope of The Cryosphere. The applied inversion scheme is at the forefront of algorithms applied in the field of hydrogeophysical inversion so far, e.g., also considering uncertainties. While being applied in hydrological research rather frequently, to the reviewer's knowledge, it is one of the first being applied in a permafrost modeling study. The research question (estimating soil organic carbon content under freeze/thaw conditions) is challenging and justifies to test the approach based on synthetic simulations. I have one major comment that I would like to see elaborated in a revised version of the manuscript. Then I am looking forward to seeing the paper published in The Cryosphere. Major comment:

While I am very excited by the inversion approach, I am missing a derivation of the relationship between OC and the measured state variables (apparent electrical resistivity, soil moisture, soil temperature) that should be the basis for a successful, hence, a related section should be added to the revised manuscript including the respective references. In that context I am also somewhat disappointed about the selection of scenarios chosen for testing the inverse parameter estimation as well as the related discussion because I am often missing the physical basis (= discussion of relationships between different parameters or parameters and models) – please see also specific comments below.

Reply: We added text describing why we can estimate OC content from soil liquid/ice content, soil temperature and resistivity at lines 7-15 page 3 in the revised version as below:

Because OC and mineral content largely influence hydrological-thermal parameters

(i.e., thermal conductivity, heat capacity, hydraulic conductivity and retention curve; see Appendix A), they are the main soil properties that control the subsurface hydrological-thermal dynamics. As a result, OC and mineral content can be potentially obtained by inverting observations of hydrological-thermal state variables (i.e., soil liquid/ice water content and soil temperature) and their correlated observables (e.g., electrical resistivity). However, so far there has been no effort using this approach to indirectly estimate these soil properties.

Specific comments:

P 1, L 2: Better use "Soil Organic Carbon Content" here.

Reply: This has been corrected.

P 1, L 17: This already relates to my major comment above but even more to the description of the physical relationships later on in the manuscript: In order to be able to estimate OC content, there must be a physical relationship between liquid water content, temperature, apparent electrical resistivity and OC which needs to be elaborated and explained clearly and which should also be the basis for defining the scenarios.

Reply: As presented in the main comment, we explain these relationships in lines 17-20 page 1, lines 7-15 page 3, line 27-28 page 4. Detailed equations concerning these relationships are presented in the Appendix A.

P 1, L 20: Please provide examples for land surface processes ("such as. . .") P 1, L 24: I would prefer to use "liquid water content and ice content" here.

Reply: Some examples of land surface processes (solar radiation balance, evapotranspiration, snow accumulation and melting) have been added to the revised manuscript (lines 21-22, page 1).

"liquid water content and ice content" have been used (line 25, page 1).

P 2, L 22-25: Please add references and values for all listed properties here. P 2, L

30: Please correct "into a land surface model"

Reply: These values come from CLM model as described by Farouki (1981). The reference has now been added to the revised manuscript (lines 27, page 2).

P 3, L 9: Please correct "used a single dataset"

Reply: Correction has been made

P 3, L 14-26: There are also numerous studies from Europe and Asia that use geophysics to study PF processes which should be honored in this short review as well.

Reply: The following paper was added to the review (lines 22-24, page 3): Schwamborn, G. J., Dix, J. K., Bull, J. M., & Rachold, V. (2002). High-resolution seismic and ground penetrating radar–geophysical profiling of a thermokarst lake in the western Lena Delta, Northern Siberia. Permafrost and Periglacial Processes, 13(4), 259-269.

P 4, L 10: Please correct "freeze-thaw", however freeze-thaw does not necessarily require the presence of snow

Reply: "snow-free" was removed. "freeze-thaw" was corrected

P 4, L 15: Which property? Please clarify.

Reply: we modified that sentence as below (lines 24-85 page 4): Building on recent advances in the use of electrical methods in the permafrost (e.g., Minsley et al., 2016; Dafflon et al., 2017) as well as coupled hydrogeophysical inversion approaches described above, this study focuses on the development of an inverse approach that uses single or multiple datasets (soil liquid/ice, soil temperature and electrical resistivity) to estimate OC content, which is a main factor that governs the subsurface hydrological-thermal dynamics

P 4, L 32-33: Here the authors refer to the dependence of apparent resistivity to ice/liquid water content and soil temperature. I completely agree. But where is OC?

Reply: Please see the explanation in the main comment. We want to clarify that OC is not directly related to the above properties through a petrophysical relationship. Instead OC influences hydro-thermal dynamics that in turn influence soil temperature and soil liquid water content. Thus, they are indirectly related. This also means that OC could not be directly derived using a petrophysical relationship only (except if we assume a strong correlation between water content or porosity and OC, for example). The challenge associated with this indirect relationship strengthens the value of using joint inversion. P 5, L 25: Here the authors use theta as variable for the parameters. In the appendix the same variable is used for soil moisture. I suggest to choose another variable for the parameters as most of the readers of TC will be used to theta as variable for soil moisture.

Reply: We used "p"to replace theta for parameter vector.

P 6, L 26: Please number figures in the order they appear in the manuscript.

Reply: Figures were reorganized

P 6, L 27: I think this should read "in the topsoil active layers".

Reply: This was corrected.

P 7, L 8 to end of section: I suggest to move this part of the section to the results section. I think the important message from this exercise is that porosity needs to be considered in the model as soon as OC and mineral content are distinguished. I suggest to draw this conclusion and design or select the scenarios accordingly.

Reply: In this section, we considered the two cases: 1) soil porosity is a function of OC and mineral content and 2) soil porosity is independent from OC and mineral content. Then, we plotted the variation of thermal conductivity, heat capacity and thermal diffusivity as a function of OC, sand content and liquid saturation for the two cases. Our key message is that when soil porosity is determined by OC and sand content, the soil thermal properties are more sensitive with the variation of OC and sand content.

Hence, we interpret the hydrological-thermal dynamics to be more sensitive to OC and sand content, and therefore, associated properties will be estimated with lower uncertainties. We tested this assumption by comparing scenario 8 and 9 (section 3.2.4). Because consideration of two cases related to the change of structure of CLM we will keep this section here.

P 7, L 24: Replace "the figure" by "Figure 2" P 7, L 29: Please correct "depend" P 7, L 30: Please correct "to a quick change"

Reply: These were corrected

P 8, L 6: The authors chose to use Archie's Law here (please add reference). I am not an expert in ERT analysis but is it also applicable for soils with high OC? If yes, the relationship to OC would be in the porosity and the soil electric conduction (OC being a volume fraction of soil matrix then). How does Archie's Law deal with ice content (reduced porosity with ice in fact being a part of the soil matrix now)? As far as I see, ice content is only considered to calculate pore water conductivity but no changes in porosity. I would like to ask the authors to elaborate on that in this section.

Reply: This is very interesting observation of the reviewer that has not been explored before to our knowledge. Yes, in this case, the ice content does not influence the Archie's model. We added below sentence to the revised version (lines 22-2427-29 page 8): It is worth noting that the reduction of porosity due to ice content in this study was not considered. How ice content influences the Archie's equation will be considered in the future research.

P 8, L 27: Does this formula also apply to organic rich soils? Pleas add a reference. P 9, L 12-16: Long sentence and difficult to understand. Can you split it into two?

Reply: P8,L27: This equation considers the relationship between temperature and electrical conductivity, so it can be used for organic soils. The reference was added.

P9, L12-16: We modified the sentence as below (lines 1-5 page 10):

The deterministic optimization algorithm was used to approximate the initial set of model parameters and initial covariance matrix of the proposal distribution for stochastic optimization. Consequently, the estimated parameters are more rapidly obtained than only using a single stochastic algorithm with arbitrary initial parameters.

P 13, L 3-6: This information is not essential for the manuscript and can be removed. (L5: Please correct "used".)

Reply: This sentence introduces the site study that we used for synthetic simulation. For clarify, we modified it as below (line 3-5 page 14): The synthetic column was developed to mimic typical soil and petrophysical properties associated with a high-centered polygon at an intensive study transect (NGEE-Artic, Barrow, Alaska) (Figure 4).

P 13, L 16: This is not so relevant for the synthetic study but probably for real-world cases to be carried out in future: As we are dealing with soil layers here, is it really reasonable to interpolate the measurements? Often we observe distinct soil horizons and also very distinct "jumps" in soil properties at layer boundaries and more constant properties along the different soil layers. Another aspect: how would ERT measurements have to be inverted considering gradients in layer properties? Please rethink whether interpolation is reasonable here.

Reply: As shown in Figure 3 and equation 19, we have distinct top and bottom layers with constant properties. We only interpolated soil properties of the layers between these two layers (from 0.015 to 1 m). We agree that various approaches could be considered for the problem of interpolation/layers. For real-world application, the choice will likely be guided by the type of observed variation.

P 13-14: Scenarios: As already stated in the general comments I am a little wondering about the chosen scenarios. It would be helpful if to each set of scenarios a short explanation would be added why exactly these scenarios were chosen. In that context, I have the following questions which I would like to ask the authors to elaborate in the revised manuscript: Where is the relationship between OC and apparent resistivity

in scenarios 1 and 2? Maybe this requires further explanation but as far as I see, in Archie's Law (eq. 4) OC primarily contributes to sigma via porosity and the soil's electrical conduction (which is fixed in this case however). So you basically vary a parameter in Archie's Law (or the BERT part of the scheme) whereas the "true" OC is modeled in CLM. How does that work? Is it reasonable to use electrical resistivity only as target variable? P 16, L 5-7: "This indicates that the apparent resistivity data is insensitive to OC content at z=0.6 m. This is reasonable, because this depth is within the permafrost (see Figure 12), where temperature insignificantly changes over time."
> so apparent electrical resistivity depends on temperature but not on OC?

Reply: The purpose of each scenario was presented in the initial manuscript. It is at lines 5-20 page 15 of the revised version.

The electrical resistivity can be used to estimate OC and sand content because it relates to soil temperature and soil liquid/ice content. Because below 0.6 m, water is frozen and temperature shows insignificant variation, the resistivity does not change over time, and therefore, it is difficult to estimate OC content below 0.6 m.

P 14, L 26-31: It would be nice to have the time series plots (cf Figure 12) already here including the apparent electrical resistivity data. Why didn't the authors decide to use the same time span for temperature and liquid water content? The changes in moisture and temperature (expect for layer 1) are rather low which I presume to be the main reason for the large uncertainties discussed later on. Hence the set of measurements is not really ideal for testing the inversion scheme. In order to obtain good parameter fits, the ranges in state variables where the model is fitted to should be large. Unfortunately this applies then to all tested scenarios.

Reply: For electrical resistivity, each time we have multiple values so it is better to show results in 1:1 plot.

The reason we did not use the same time span for temperature and soil moisture is that during the winter season, soil temperature still show relatively large variation so it

contains information for parameter estimation. Hence, we selected temperature data in both winter and summer season. However, soil liquid is nearly equal to zero over the winter so it does not contain any information for parameter estimation. Therefore, we only selected soil moisture data in summer season to reduce the computation time.

P 14, L 26: I guess, this should be figure 11?

Reply: This sentence is about the procedure to perform synthetic simulation so it should be figure 5.

P 15, L 8: please correct: "uniformly" P 15, L 15: Please correct "...8 and 9 is larger..." P 15, L 20: Please correct: "influence of measurement error" P 15, L 21: delete "of" P 16, L 21: Please correct: "jointly"

Reply: These errors were corrected

P 16, L 23-25: "These synthetic experiments suggest that given this depth is located within the permafrost (see Figure 12), the apparent resistivity, liquid content and temperature data are insensitive to OC content." Why? Please explain. I presume the uncertainties are so large because there are almost no changes in state variables the model is fitted to.

Reply: Yes, our explanation is shown in lines 8-10, page 18 as below: This is because within the permafrost, the soil temperature and ice/liquid content exhibit much smaller variations than in active layer, in both time and space.

P17, L 9-17: I do not understand Figure 9. Why is reasonable to test correlations between all estimated parameters? What determines when a parameter is reliably estimated?

Reply: Figure 9 is plotted to observe the parameter variation and correlation between estimated parameters. A parameter is more reliable if it has narrow variation range and is independent from other parameters.

P17, L 29: remove "are" P 17, L 31: So do "thermal parameters" mean ice content here? P 18, L 18: Please correct "confidence interval" P 18, L 26 to end of paragraph: Figure 12 is missing in manuscript. P 19, L 30: Please correct "in a 1-D soil column"

Reply: These were corrected.

"thermal parameters" in this case mean thermal conductivity and heat capacity. We added these explanations in the revised version.

P 20, L 5-7: Again: Here the authors relate the large uncertainty to the missing range in tempera-ture and moisture. However, again: Which are the properties that are related to OC? It's neither directly temperature nor moisture.

Reply: The OC determines the thermal conductivity and heat capacity and thus governs the soil moisture and temperature dynamics.

P 20, L 13-15: Also here a more detailed conclusion would be helpful: In which way does the joint inversion help to constrain the model? How to the various measurements and models contribute here?

Reply: When we have more measurement data, we have more information about hydrological-thermal dynamics. For example, we only have soil moisture and soil temperature at a few locations. When ERT data are available, it supplements information in the whole soil profile. However, compared to soil moisture and temperature measurement, ERT data show more limited spatial resolution. So by combining them together we can better constrain the inversion.

How to the various measurements and models contribute here? These measurements can contribute to the inversion via the posterior distribution formula. We added explanations in line 28 page 10 – line 4 page 11 as below:

Intuitively, $\sigma\_i\hat{}2$ works as an inverse weighted factor of contribution of measurement $(y\_i)$ ÌĆ to the posterior distribution $p(Y|p)$. A measurement with a higher variance of measurement error has a smaller contribution to construct the parameter posterior

distribution. In addition, for joint inversion, $\sigma\_i\hat{}2$ helps to removes the influence of measurement units of different data types.

P 20, L 17-19: ... and the small range in "measured" soil moisture states. P 21, L 1-2: This information is not relevant for the study and should be removed. P 21, L 4-11: Also not relevant here – please remove. References: Please check references for typos, also check that the European "Um- laute" are in-cluded in names (e.g. Etzelmüller)

Reply: These were corrected

P 22, L 13: only cite papers which are at least accepted Appendix A: Definitions for most oft he parameters are missing.

Reply: The cited paper was accepted and in press. The definitions of parameters in appendix A were added in the revised

Please also note the supplement to this comment:
http://www.the-cryosphere-discuss.net/tc-2017-1/tc-2017-1-AC1-supplement.pdf

---

## Author Comment (AC2) · 14 May 2017

We would like to thank the reviewer for their evaluation and constructive comments, which definitely helped to improve our paper.

I enjoyed reading your paper and the authors of the paper describe an interesting combination of coupled hydrogeophysical inversion using also thermal properties. The investigation of uncertainties and to analyze the influence of different data sets to im-

prove the results is a very interesting topic and highly important for coupled inversions. Coupled hydrogeophysical inversions have been widely used in the last years, but the extension to thermal parameters is still rare. Generally, I think the paper is well written and all the important steps are nicely explained. I have some minor comments that could help to make same things clearer and improve the manuscript. After addressing them I recommend publication.

1. Introduction: Please discuss in more detail coupled hydrogeophysical inversion in terms of other geophysical methods. What are the benefits of ERT compare to GPR or seismic and why did you prefer this for you study. Mention the resolution of different methods and what are the limitations of what can be obtained.

Reply: We added text describing why we chose to develop the hydrogeophysical inversion approach using ERT in the revised version as below (lines 8-14, page 4):

Of the geophysical techniques commonly used for monitoring the shallow subsurface, ERT is increasingly common because it can autonomously provide 2- or 3-D time-lapse measurements with a relatively high spatial resolution, is sensitive to properties influencing hydrological-thermal dynamics, and is particularly suitable for field deployment over a long period of time. As a result, we use ERT data in this study.

2. Regarding the ERT data: a. Page 13: It would be nice to show also one ERT transect from the measured data and indicate the defined boundaries and structures in there. b. Please give more information of the ERT and the inversion. How reliable are the ERT results at a depth of 0.1m when using a spacing of 0.5m? c. Page 14, last paragraph: Considering measurement errors are highly important, but did you also consider uncertainties of the actually layer thicknesses obtained by the ERT?

Reply: a) Figure 4 was modified to add an inversion of ERT data collected at the Barrow, AK site on August 2013. The permafrost and active layers shown in the electrical resistivity map are discussed in the caption based on their resistivity values.

b) As shown in Figure 4, ERT results at depth of 0.1 m are not clearly distinguished. That is the reason why joint inversion of ERT with soil liquid and soil temperature helps to improve the parameter estimation.

c) In coupled hydrogeophysical inversion, subsurface state variables (soil liquid/ice content and temperature) within the hydrological computation domain are transformed into resistivity values within the BERT computational domain. After that, the forward BERT model is used to simulate ERT data (resistance and inferred apparent resistivity). The simulated apparent resistivity data are then compared with the observed ERT data (apparent resistivity) (see procedure in Figure 1). We do not perform BERT inversion so we avoid the errors associated with geophysical inversion.

d. Page 15: Why do you use just 7 data set of the ERT, when you have data available for every day?

Reply: We only selected the 7 ERT datasets that represent the most important events. Adding more ERT data takes a longer time for the inversion but does not provide much more information about system dynamics. Because additional ERT does not significantly improve the parameter estimation, we did not include them. Still, acquiring ERT data with higher temporal resolution than used in this study is suggested to understand and select the datasets related to strong changes in soil resistivity (no necessarily known in advance).

Technical corrections: The manuscript is very detailed, which is generally very good, but please try to remove unnecessary sentences to shorten the text and to better concentrate on the results. Avoid sentences like "Figure XX shows. . ..". I listed here some examples that could shorten the text. Please check this for all the location where figures and tables are introduced. For example: e. Page 17: last paragraph. The three sentences can easily be combined to one. f. Page 18 starting line 4: First sentence not necessary and combine with second sentence. g. Page 19, second paragraph: Rewrite to "The comparison between synthetic and predicted apparent resistivity data

(Figure 14) shows that there is a very good agreement between them with no bias. . ...

Congratulations on a very nice job!

Reply: Some unnecessary sentences were removed/combine as suggested by the reviewer.

Please also note the supplement to this comment:
http://www.the-cryosphere-discuss.net/tc-2017-1/tc-2017-1-AC2-supplement.pdf
* * *

---

## Author Response (AR2)

*We would like to thank the Editor and two reviewers for their constructive comments, which definitely helped to improve our paper.*

**Review 1:**

Dear authors,

thanks for considering the requested changes in the manuscript and by updating it.
I think the paper is close to be ready for publication. I just have one comment left that should be addressed and changed before.

Generally, I'm missing references in the introduction and methodology part. Especially references that are less related to the authors and their work.
For example, several other research groups work since years on joined and coupled inversion approaches maybe not in the permafrost area but in terms of hydrology and geophysics using similar approaches as used in this manuscript. Also several publications are available of coupled hydrological inversion using different geophysics methods. Please add references and honor also the work of other research groups more.

*Reply: Based on suggestion of the reviewer, several coupled hydrogeophysical studies below were added to the reference. (lines 7-9 page 4)*

1. *Busch, S., Weihermüller, L., Huisman, J. A., Steelman, C. M., Endres, A. L., Vereecken, H., & Kruk, J. (2013). Coupled hydrogeophysical inversion of time-lapse surface GPR data to estimate hydraulic properties of a layered subsurface. Water Resources Research, 49(12), 8480-8494.*

2. *Camporese, M., Cassiani, G., Deiana, R., Salandin, P., & Binley, A. (2015). Coupled and uncoupled hydrogeophysical inversions using ensemble Kalman filter assimilation of ERT-monitored tracer test data. Water Resources Research, 51(5), 3277-3291.*

3. *Herckenrath, D., Fiandaca, G., Auken, E., & Bauer-Gottwein, P. (2013). Sequential and joint hydrogeophysical inversion using a field-scale groundwater model with ERT and TDEM data. Hydrology and Earth System Sciences, 17(10), 4043-4060.*

4. *Huisman, J. A., Rings, J., Vrugt, J. A., Sorg, J., & Vereecken, H. (2010). Hydraulic properties of a model dike from coupled Bayesian and multi-criteria hydrogeophysical inversion. Journal of Hydrology, 380(1), 62-73.*

5. *Irving, J., & Singha, K. (2010). Stochastic inversion of tracer test and electrical geophysical data to estimate hydraulic conductivities. Water Resources Research, 46(11).*

6. *Pollock, D., & Cirpka, O. A. (2012). Fully coupled hydrogeophysical inversion of a laboratory salt tracer experiment monitored by electrical resistivity tomography. Water Resources Research, 48(1).*

**Reviewer 2**

I have also reviewed the first version of the manuscript and the authors have revised most of my comments.

I have still some reservations regarding some parts of the manuscript where the authors are urgently requested to be formally correct in their wording: As the authors state clearly in their response to my first review, there is no direct relationship between OC and electrical resistivity. In their conclusions of the revised version, they again state (P. 22, L 16-17) „we concentrated on the impact of OC content on the soil electrical resistivity via its hydrological-thermal properties". Formally and physically correct, only properties or states that influence electrical conduction can influence measured electrical resistivity. The most important ones are clay content, dissolved ions in the water phase in combination with the continuity of the water phase which allows electrical conduction, and temperature. Most other properties only have indirect effects on electrical resistivity as they are mostly (often non-linearly) related to one (or even some at the same time) of the properties stated above. I know that there are many hydrogeophysical applications that follow a similar approach, however, it is important to stick to the physical basis of the processes. Consequently, the wording needs to be carefully revised in this respect.

*Reply: The foundation behind the applications of hydrogeophysical inversion is that ions are dissolved in water and move with water. Therefore, geophysical techniques that measure soil electrical conductivity can be used to track the water dynamics. In addition, with the same amount of ions in water, if water reduces for some reasons (for example, by freezing), the ion concentration (and correlated water electrical conductivity) will increase. By contrast, when soil temperature reduces, ion mobility reduces, which decreases water conductivity. Because OC influences the dynamics of both temperature and soil moisture, it influences the water electrical conductivity. We agree with the reviewer that OC indirectly influences soil electrical conductivity. To emphasize the indirect impact of OC on water electrical conductivity via soil water and temperature, we modified the sentence in P22, L16-17 as below:*

*In this study, we concentrated on the indirect impact of the OC content on water electrical resistivity via soil water and temperature. (Line 16-17 page 22)*

Besides all the challenges at unfrozen conditions, this is particularly important as soon as the soil becomes frozen and, hence, the water phase becomes discontinuous causing the characteristic drop in electrical resistivity during freezing. This effect is not considered by the empirical Archie's Law. I did a quick literature scan mayself and found some work by Hauck and collaborators who did some nice work employing ERT in periglacial and glacial environments. So the authors may take a look at these publications. For this manuscript, it should be ok to include a review about application of Archie's Law in permafrost research which also mentions the difficulty with using this equation and

discusses the limits of the approach followed in this study. Formalizing this process correctly, which would mean finding a better equation than Archie's Law, could be one grand challenge in future research to really establish ERT for quantitatively investigating freeze/thaw and related processes

*Reply: Thank to reviewer for referring the research of Hauk and collaborators. Regarding petrophysical relationship in cold region, Hauk et al. (2011) developed a 4-phase soil model (consist of soil matrix, ice, liquid and air) and used it to estimate ice and liquid content from combined ERT and seismic measurements. Similar to our study, they used Archie's model to link soil water liquid with soil electrical resistivity, which is obtained from ERT measurement. The other components (ice, soil matrix and air) were linked to P-wave velocity, which is obtained from seismic measurement. This is an interesting approach to jointly combine ERT and seismic data for estimating ice and water liquid. A review about the works of Christian Hauk was added to the introduction part as below:*

*Hauck et al. (2011) developed a 4-phase model of soil matrix, ice, liquid and air and used it to estimate soil liquid and ice content from combined ERT and seismic measurements in the Swiss Alps. (lines 25-27, page 3)*

*Different from Hauk's studies, we used multiple time-lapse ERT measurements but we did not have seismic data in this study. So we cannot use 4-phase model. We will explore the possibility to integrate Hauk's petrophysical model into our coupled hydrogeophysical inversion when both electrical resistivity and seismic (or GPR) measurements are available. It is worth noting that in 4-phase soil model of Hauk, he assumed that the soil porosity and Archie's model parameters are known. In hydrogeophysical inversion, we can set them as unknown parameters and estimate them (like we did in this study). In addition, in Hauk's study, they only used single ERT and seismic measurements at a time instant. In hydrogeophysical inversion, we will use multiple time-lapse measurements. As a result, the inversion is potentially better constrained. We added the following sentences to the revised manuscript:*

[revised manuscript text omitted]